# Dynamic Subgoal-based Exploration via Bayesian Optimization

**Yijia Wang**                                                                *yiw94@pitt.edu*
*University of Pittsburgh*

**Matthias Poloczek**\*                                          *matthias.poloczek@gmx.de*
*Amazon*

**Daniel R. Jiang**                                                      *drjiang@meta.com*
*Meta AI, University of Pittsburgh*

**Reviewed on OpenReview:** *https://openreview.net/forum?id=ThJl4d5JRg*

## Abstract

Reinforcement learning in sparse-reward navigation environments with expensive and limited interactions is challenging and poses a need for effective exploration. Motivated by complex navigation tasks that require real-world training (when cheap simulators are not available), we consider an agent that faces an unknown distribution of environments and must decide on an exploration strategy. It may leverage a series of training environments to improve its policy before it is evaluated in a test environment drawn from the same environment distribution. Most existing approaches focus on fixed exploration strategies, while the few that view exploration as a meta-optimization problem tend to ignore the need for *cost-efficient* exploration. We propose a cost-aware Bayesian optimization approach that efficiently searches over a class of dynamic subgoal-based exploration strategies. The algorithm adjusts a variety of levers — the locations of the subgoals, the length of each episode, and the number of replications per trial — in order to overcome the challenges of sparse rewards, expensive interactions, and noise. An experimental evaluation demonstrates that the new approach outperforms existing baselines across a number of problem domains. We also provide a theoretical foundation and prove that the method asymptotically identifies a near-optimal subgoal design.

## 1 Introduction

Reinforcement learning (RL) is becoming the standard for approaching control problems – usually modeled by a Markov decision process (MDP) – in environments whose dynamics are unknown and learned from data. In many applications involving navigation tasks, rewards are sparse and delayed. Since most RL algorithms rely, at least initially, on random exploration, this can cause an agent to require a large, often impractical number of interactions with the environment before obtaining any rewards. Simultaneously, in real-world settings, it is often the case that fast and cheap interactions with the environment are not available, making it nearly impossible to apply RL algorithms. To address the two issues of sparse rewards and expensive interactions in navigation tasks, our objective in this paper is to design methods for learning better exploration policies in a *cost-efficient* manner: specifically, we propose a Bayesian optimization approach to optimize an exploration strategy based on *subgoals*, where each *subgoal* is defined as a set of states that the agent must reach, serving as an intermediate target for the agent to "complete" before navigating to the primary goal.

An illustrative example comes from the field of robotics: autonomous systems have long been used to explore unknown or dangerous terrains (Matthies et al., 1995; Apostolopoulos et al., 2001; Ferguson et al., 2004;

---

\*The work was done before Matthias joined Amazon.

Thrun et al., 2004). Policies learned offline (e.g., via a simulator) are common in these situations, but it may be beneficial to introduce agents that execute an offline-learned exploration policy to guide the learning of an *online* policy that can better tailor to the details of the test environment. An example of this general idea can be found in Matthies et al. (1995), which describes the design of a rover for the Mars Pathfinder mission. One of the main tasks is navigating the rover in a rocky terrain and reaching a goal (the test environment). To train for the eventual mission, the engineers utilized an "indoor arena" that mimics the test environment. The need for cost-efficient training also arises in the setting of safe robot navigation (Oliveira et al., 2020). Existing approaches to exploration have largely ignored the need to be cost-efficient during the training process and therefore are challenging to apply to real-world scenarios (see Section 2 for a detailed discussion of related work).

In our setup, an agent is given a fixed (and small) number of opportunities to train in environments randomly drawn from a distribution $\Xi$ (henceforth, we refer to these as "training environments"), with the caveat that each interaction in the training environment *incurs a cost*. After these opportunities are exhausted, the agent enters a random *test environment* $\xi \sim \Xi$ and executes an underlying RL algorithm to adapt to the particulars of $\xi$, while aided by the higher-level exploration strategy learned for $\Xi$. One can view this formulation as a meta-optimization problem with two levels: an upper-level problem to select an exploration strategy, represented by parameters $\theta$, and a lower-level RL task that explores with the help of the exploration strategy $\theta$ on an environment instance $\xi \sim \Xi$.

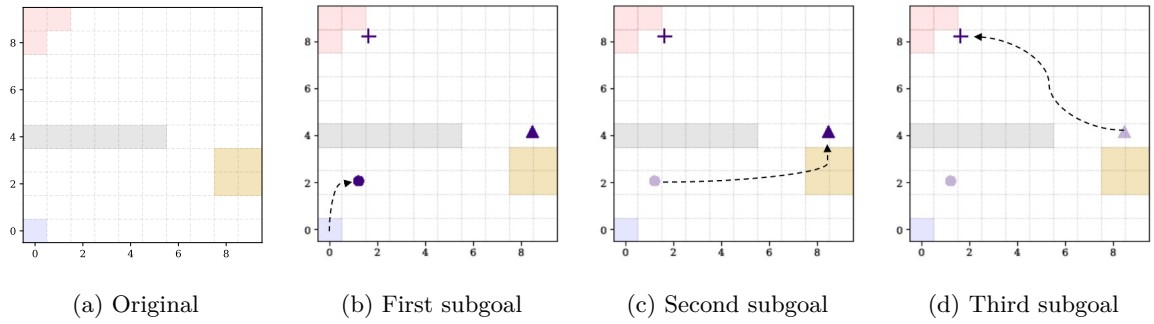

(a) Original      (b) First subgoal      (c) Second subgoal      (d) Third subgoal

Figure 1: Example of a dynamic subgoal exploration strategy. The first, second, and third subgoals are denoted by the circle, triangle, and cross, respectively. The blue square is the starting location of the agent, the grey region is a wall, the yellow region is the location of the key, and the red region is the door (goal). Note that although the second subgoal is not exactly in the location of the key, it brings the agent to the correct vicinity, allowing the underlying RL algorithm (executed online) to further adapt to the environment's particular details.

We propose optimizing over a class of *dynamic subgoal exploration strategies* in the upper-level optimization problem. To illustrate this concept, consider the sparse-reward environment shown in Figure 1a, where an agent is tasked with picking up a "key" in the yellow region, in order to exit the "door" in the red region. The grey region is a wall. An RL algorithm paired with a naive exploration strategy making use of random actions (such as $\epsilon$-greedy) requires a prohibitively large number of random actions before finding a suitable path to the door through the key, while avoiding the wall. A dynamic subgoal strategy is an *ordered* set of subgoals (along with associated rewards leading to each subgoal, omitted here for illustrative clarity) that leads the agent on a trajectory where the underlying RL algorithm is *more likely to discover* the optimal behavior. Figures 1b-1d together show an example of a dynamic subgoal exploration with three subgoals, which first leads the agent to the vicinity of the key and later towards the door. Note that the situation here in Figure 1 is simplified in that we are actually interested in finding dynamic subgoal strategies that work on average across a distribution of environments, rather than a single environment.

## 1.1 Our Contributions

Our main contributions are as follows. We first propose a framework for *cost-efficient learning* of a dynamic subgoal exploration strategy for a distribution of environments; in other words, interactions with the environment are expensive *during training*, making most gradient-based approaches infeasible. We instead

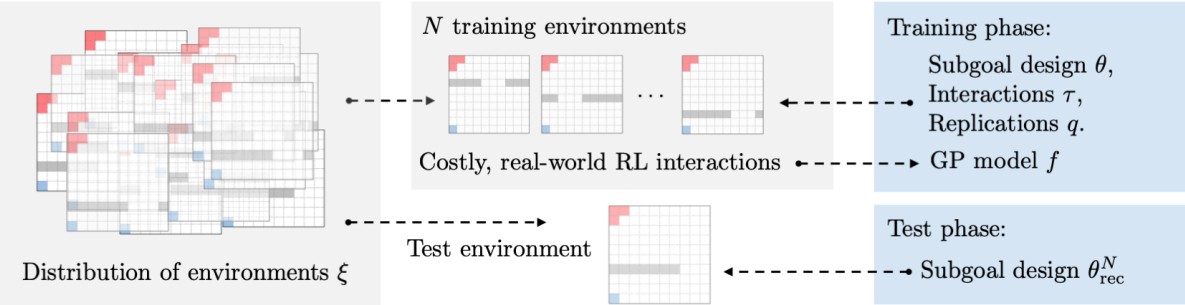

Figure 2: Outline of the BESD algorithm. During the training phase BESD optimizes an exploration strategy (represented as subgoals) on sampled training environments. It then utilizes the learned subgoal design as an exploration strategy in the test environment to train an effective policy within a limited number of interactions.

leverage the Bayesian optimization (BO) paradigm, a well-known class of sample-efficient optimization techniques (Brochu et al., 2010; Snoek et al., 2012; Herbol et al., 2018; Frazier, 2018), and propose a new acquisition function as a core ingredient of our approach. The Gaussian process (GP) surrogate model used by the BO formulation has the ability to reason about the *learning curve* of the underlying RL algorithm, enabling us to introduce two additional levers in the BO learning process to improve cost-efficiency: (1) how long to run each episode of training, (2) the number of replications to run in each training environment. These levers allow us to intelligently trade-off running a longer trial versus more replications of shorter trials; the motivation is that, given $\tau_1 < \tau_2$, an accurate evaluation of a particular exploration strategy $\theta$ after $\tau_1$ steps may be more informative than a noisy evaluation of $\theta$ after $\tau_2$ steps, even though the same number of environment interactions are used in both cases. The proposed algorithm, *Bayesian exploratory subgoal design* (BESD), is outlined in Figure 2. We also prove an asymptotic guarantee on the quality of the solution found by our approach, compared to the best possible subgoal-based exploration strategy within a given parameterized class.

## 2 Related Work

Our framework of cost-efficient learning of exploration strategies through BO appears to be distinct from existing formulations in its strong focus on expensive environmental interactions *during training*, made possible through the additional control levers of episode length and number of replications. Nevertheless, our work is related to a number of distinct areas of study: Bayesian optimization, exploration for RL, intrinsic reward and reward design in RL, multi-task RL, and transfer learning. Here, we attempt to give a tour through the various strands of relevance in each field.

### 2.1 Bayesian Optimization

BO is a technique for optimizing black-box functions in a sample-efficient manner, in particular for tuning ML models and design of experiments (Snoek et al., 2012; Brochu et al., 2010; Frazier, 2018; Herbol et al., 2018). BO methods for problems with multiple information sources or fidelities (Swersky et al., 2013; 2014; Feurer et al., 2015; Domhan et al., 2015; Li et al., 2017) is especially relevant to our proposed method's ability to select the length of an RL training episode, which builds upon ideas from Picheny & Ginsbourger (2013), Poloczek et al. (2017), and Klein et al. (2017). The first paper proposes fitting GP to partially converged simulations, and the latter two propose acquisition functions that consider the ratio of "information gain" to cost of evaluation. Our approach also reasons about multiple replications in an environment, similar to the problem studied in Binois et al. (2019) in the context of computer experiments. Our work fills a gap in the BO literature where the length of training and number of replications are *selected jointly* in a cost-aware setting, a natural and powerful idea that has not been considered in the literature. Our theoretical analysis builds upon techniques developed in Frazier et al. (2008) and Poloczek et al. (2017) but extend them in new

directions, accounting for the ability to select the number of replications, and providing a characterization of the asymptotic suboptimality due to using a discretized domain.[1]

BO has previously been applied in the setting of navigation planning. Martinez-Cantin et al. (2007), Martinez-Cantin et al. (2009), and Binney et al. (2013) use BO to optimize a sequence of waypoints for a robot to follow. While our method similarly optimizes a sequence of subgoals, we use the subgoals as an exploration strategy (over a distribution of environments) on top of an existing RL algorithm, rather than as a *direct specification* of the control policy. In order to allow subgoals to provide exploration in a "plug-and-play" manner for existing RL algorithms, our approach also features a novel integration of subgoals with potential-based intrinsic rewards.

In two other works, Tesch et al. (2011) and Garcia-Barcos & Martinez-Cantin (2021), BO is directly applied to optimize a parameterized policy, but this is limited to low-dimensional parameterizations of the policy: Tesch et al. (2011) tune a two-dimensional gait parameter, while Garcia-Barcos & Martinez-Cantin (2021) tune four policy parameters. In our work, we augment an underlying RL algorithm that can learn arbitrary policies with a BO-optimized low-dimensional exploration strategy, striking a balance between flexibility and cost-efficiency.

## 2.2 Exploration in Reinforcement Learning

Naive exploration strategies such as $\epsilon$-greedy can lead to unreasonably large data requirements, making exploration a commonly studied topic in RL. Most existing work focus on proposing a fixed exploration strategy that is executed for a single underlying environment. For example, some previous related work employ approaches based on optimism (Kearns & Singh, 2002; Stadie et al., 2015; Bellemare et al., 2016; Tang et al., 2017) and posterior sampling (Osband et al., 2016; Russo & Van Roy, 2014; Osband & Van Roy, 2017; Morere & Ramos, 2018) to guide exploration. Others insert an active learning (Shyam et al., 2019) or experimental design (Mehta et al., 2021) perspective into the model-based RL framework.

Our work departs from these existing studies in that we formulate the problem of exploration as a meta-optimization over a parameterized class of exploration strategies and aim to find a suitable strategy for a distribution of environments. A more closely related paper is Gupta et al. (2018), which extends the model-agnostic meta-learning (MAML) approach of (Finn et al., 2017a) to the problem of exploration for a set of tasks in a way that is similar in spirit to our formulation. However, their gradient-based approach is not sample-efficient and they do not consider costly environment interactions during training. In addition, Gupta et al. (2018) make use of task-specific parameters during training, limiting their approach to a small set of environments. For a more comprehensive list of methods for exploration in RL, we refer the reader to the excellent survey of Amin et al. (2021).

## 2.3 Hierarchical Reinforcement Learning and Options

Our proposed approach is related to the hierarchical reinforcement learning (HRL) framework, which refers to methods that decompose a complex, long-horizon problem into smaller subtasks; see Barto & Mahadevan (2003) and Pateria et al. (2021) for extensive reviews of the topic. A well-known type of HRL is *feudal reinforcement learning*, introduced in Dayan & Hinton (1992), where a high-level manager delegates low-level workers to complete subtasks. Examples of more recent work that follow this feudal hierarchy paradigm include Kulkarni et al. (2016), Levy et al. (2018), and Nachum et al. (2018). Our work exhibits a similar flavor in that a high-level BO method sets a subgoal-based exploration strategy, which is then executed by the underlying RL algorithm.

The concept of options, which are temporally extended actions represented as a policy and a termination condition, also fall under the HRL framework. Options can improve the efficiency of RL through the use of previously acquired "skills" (Sutton et al., 1999; Precup et al., 1998). These skills might be acquired with the help of a human, either fully user-specified (e.g., Jothimurugan et al. (2021)) or obtained from expert demon-

---

[1]Discretizing the domain is a common computational technique used when optimizing complex acquisition functions, but we improve upon the existing theoretical analysis in Poloczek et al. (2017) with an explicit characterization of the induced error.

strations (e.g., Pan et al. (2018), Paul et al. (2019)). In this paper, a subgoal is a particular type of option and therefore, our dynamic subgoal exploration strategy can be thought of, at a high level, as a sequence of options.

Of particular relevance to our work is when options are automatically discovered, a problem that is well-known to be challenging. One stream of work views option discovery to be (at least somewhat) detached from the RL reward maximization objective, using state visitation frequencies (Stolle & Precup, 2002; McGovern & Barto, 2001; Goel & Huber, 2003), clustering (Mannor et al., 2004), novelty (Şimşek & Barto, 2004), local graph partitioning (Şimşek et al., 2005), or diversity objectives (Eysenbach et al., 2018; Zhang et al., 2020), to name a few examples. Approaches that considers a joint objective for option learning RL reward maximization objective like ours (Kulkarni et al., 2016; Vezhnevets et al., 2016; Bacon et al., 2017; Frans et al., 2018; Veeriah et al., 2021) typically use large, neural network-based representations along with gradient-based (meta-)optimization and do not focus on cost-aware training. The method that we propose in this paper is unique from previous works in that (1) it is designed specifically for the case where cost-aware training is warranted and uses BO for option-learning, (2) it offers an integrated objective for subgoal-design and RL reward maximization, and (3) it uses a novel combination of subgoals and reward shaping, which has a simpler representation than a generic option.

### 2.4 Intrinsic Reward and Reward Design

When a particular subgoal of our proposed dynamic subgoal exploration strategy is active, we "turn on" a set of artificial rewards that incentivize the agent to move toward that subgoal (these rewards are then removed after the agent moves on to the next subgoal). Hence, the literature on intrinsic reward and reward design in RL are also relevant. Intrinsic reward (also called *intrinsic motivation*) helps an agent learn increasingly complex behavior in a self-motivated way (Randløv & Alstrøm, 1998; Ng et al., 1999; Huang & Weng, 2002; Kaplan & Oudeyer, 2004; Şimşek & Barto, 2006; Tenorio-Gonzalez et al., 2010; Pathak et al., 2017; Achiam & Sastry, 2017; Lample & Chaplot, 2017). Several works from the *reward design* literature are most closely related to our paper. Sorg et al. (2010) and Guo et al. (2016) directly optimize the intrinsic reward parameters, via gradient ascent, to maximize the outcome of the learning process. Similarly, Zheng et al. (2018) use intrinsic rewards in policy gradient, and treat the parameters of policy as a function of the parameters of intrinsic rewards. Again, these methods differ from ours in that they do not consider the costliness of training and focus on finding intrinsic rewards for a single MDP.

### 2.5 Multi-task RL and Transfer Learning

Also related to our setting are methods that aim to train agents with the capability of solving (or adapting to) multiple sequential decision making tasks (Pickett & Barto, 2002; Konidaris & Barto, 2006; Wilson et al., 2007; Fernández et al., 2010; Deisenroth et al., 2014; Doshi-Velez & Konidaris, 2016; Finn et al., 2017a;b; Pinto & Gupta, 2017; Espeholt et al., 2018; Hessel et al., 2019; Vithayathil Varghese & Mahmoud, 2020); such methods generally fall under the umbrella of *multi-task RL* or *transfer learning*. As before, many of these methods require the training of large neural networks and are not designed for a cost-aware setting. Despite their stated purpose of being sample-efficient in adapting to new tasks, most multi-task RL or transfer learning approaches do not place a strong emphasis on cost-efficiency of training on existing tasks. This is an important distinction to our work. The two papers that are closest in spirit to our work are Pickett & Barto (2002), where macro-actions are extracted from previous tasks, and Konidaris & Barto (2006), where shaped rewards are learned for a set of tasks. One drawback of Pickett & Barto (2002) is that it assumes access to optimal policies for an initial set of MDPs. Konidaris & Barto (2006) directly uses previous value functions as shaped rewards (thereby requiring the agent to solve some tasks from scratch) and does not provide an avenue for cost-effective exploration.

## 3 Problem Formulation

This section formulates the problem mathematically, by defining the original (sparse-reward) MDPs and how a dynamic subgoal exploration strategy induces an auxiliary, "subgoal-augmented" MDPs. We then describe the iterative training process.

### 3.1 Original MDPs $\mathcal{M}_\xi$ with Sparse Rewards

Consider a family of MDPs $\{\mathcal{M}_\xi = \langle \mathcal{S}, \mathcal{A}, T_\xi, R_\xi, \gamma \rangle\}_\xi$ parameterized by a random variable $\xi \sim \Xi$, where $\mathcal{S}$ and $\mathcal{A}$ are the state and action spaces, $T_\xi$ is the transition matrix, $R_\xi : \mathcal{S} \times \mathcal{A} \times \mathcal{S} \to \mathbb{R}$ is the extrinsic[2] reward function, $\gamma \in [0,1]$ is the discount factor[3], and $\Xi$ is the environment distribution (not assumed to be known, nor does it need to be finite or discrete).[4] A *sparse-reward environment* is an environment where $R_\xi$ is non-zero only for a small number of "goal" states. To ensure that all quantities are well-defined, we assume that $R_\xi$ is bounded, as is common in the reinforcement learning literature. We assume common state and action spaces across the distribution of MDPs (i.e., they are independent of $\xi$), while the reward and transition functions vary with $\xi$.

Given $\mathcal{S}$ and $\mathcal{A}$, a *policy* $\pi$ is a mapping such that $\pi(\cdot \,|\, s)$ is a distribution over $\mathcal{A}$ for any state $s \in \mathcal{S}$. For any $\xi \sim \Xi$, define the *value function* of policy $\pi$ at any state $s$ as

$$V_\xi^\pi(s) = \mathbb{E}\left[ \sum_{t=0}^\infty \gamma^t R_\xi(s_t, a_t, s_{t+1}) \,\Big|\, \pi, s \right], \tag{1}$$

where the notation of "conditioning" on $\pi$ and $s$ indicates that $s_0 = s$ is the initial state and $a_t \sim \pi(\cdot \,|\, s_t)$. For the MDP $\mathcal{M}_\xi$, its optimal value function and associated optimal policy are

$$V_\xi^*(s) = \sup_\pi V_\xi^\pi(s) \quad \text{and} \quad \pi_\xi^*(s) \in \arg\max_{a \in \mathcal{A}} \mathbb{E}\big[ R_\xi(s, a, s') + \gamma V_\xi^*(s') \,|\, s, a \big].$$

Now that we have defined the value function, let us comment on the environment distribution $\Xi$. In Section 3.4, we will formulate the meta-optimization problem, which requires that the expected performance of any policy $\pi$ over the environment distribution, i.e., $\mathbb{E}_\xi[V_\xi^\pi(s)]$, is well-defined. However, since we assumed bounded rewards, implying bounded performance $V_\xi^\pi(s)$, it will always be the case that this expectation exists and we do not require further assumptions on $\Xi$.

When the extrinsic reward function $R_\xi$ is sparse, it produces little to no learning signal for the agent. Under most RL algorithms, the agent essentially performs random exploration and does not start learning until the first time it wanders to the goal. The time it takes to find the goal under a random exploration strategy is often prohibitively long. The $\epsilon$-greedy exploration strategy, which takes a random action with probability $\epsilon$ and the best action under the current value function approximation, is an example of a random exploration strategy.

### 3.2 Dynamic Subgoal Exploration Strategies

An *intrinsic reward* is an artificial reward signal experienced by the agent that does not come directly from the environment. A *subgoal* is defined by a (usually small) set of states, such that when the agent lands in any of them, the subgoal is considered "completed." A *dynamic subgoal exploration strategy* is a sequence of subgoals, along with an associated reward shaping function for each subgoal, that provides an intrinsic reward signal for the agent. If the locations of the subgoals are chosen well, this strategy can help the agent explore the environment efficiently. We call this a *dynamic* strategy because the subgoals are turned on one-by-one and consequently introduces a new state into the MDP (described in detail below).

Suppose there are $K$ subgoals and let $\theta \in \Theta$ be a parameter that fully describes a subgoal exploration strategy, including the subgoal locations, associated rewards, and sequencing. Let $\mathcal{G}_{\theta,k} \subseteq \mathcal{S}$ be a set of "target" states associated with the $k$th subgoal, for $k \in \{1, 2, \ldots, K\}$, in the sense that if the agent lands in some state in $\mathcal{G}_{\theta,k}$, then the $k$th subgoal is considered "completed." In addition, we define an artificial reward function $g_{\theta,k}(s, s')$ that, when activated, provides a sequence of rewards that leads the agent toward subgoal $k$. Concretely, we use potential-based reward shaping from Ng et al. (1999) to achieve this. Let $\Phi_{\theta,k}$ be a *potential function*, a function that assigns a value for each state in $\mathcal{S}$, with higher potential indicating a more

---

[2]In Section 3.3, we describe how a dynamic subgoal exploration strategy supplements the extrinsic reward function with additional *intrinsic* rewards.

[3]We allow for the case of episodic MDPs, where $\gamma = 1$, provided that any policy will reach a *terminal state* with probability one. A terminal state is absorbing and any action taken in that state gives zero reward.

[4]Note that our approach also applies to the case of a *single environment* if the distribution contains only one environment.

"valuable" state. $\Phi_{\theta,k}$ should have the property that target states in $\mathcal{G}_{\theta,k}$ have the highest potential. Then, let

$$g_{\theta,k}(s, s') = \gamma \Phi_{\theta,k}(s') - \Phi_{\theta,k}(s), \tag{2}$$

for all $s, s' \in \mathcal{S}$. The definition of $g_{\theta,k}(s, s')$ in (2) can be interpreted as the difference in potential between states $s'$ and $s$ (with discount $\gamma$). This potential difference motivates the agent to move towards the target states (high potential) of $k$th subgoal. Thus, a parameterization of a set of $K$ subgoals, which forms our exploration strategy, is fully described by

$$\left( \{\mathcal{G}_{\theta,k}\}_{k=1}^K, \{g_{\theta,j}\}_{k=1}^K \right),$$

the locations and associated reward shaping functions.

**Example 1 (Key and Door Environment)** *Let us consider a distribution of maze MDPs with states $\{(i, j)\}_{1 \leq i,j \leq 10}$ and a sparse reward in the upper left corner at $(0, 10)$. In addition, suppose that the agent needs to pick up a key in order to receive the reward at $(0, 10)$, where the location of the key is uncertain but likely to be in the* right half of the room*. The environment illustrated in Figure 1 can be considered to be one possible realization from this distribution of mazes. Now, let us consider a subgoal design with $K = 3$ subgoals. The simple parameterization $\theta = (i_1, j_1, i_2, j_2, i_3, j_3)$, with*

$$\mathcal{G}_{\theta,k} = \{(i_k, j_k)\} \quad and \quad \Phi_{\theta,k}(s) = e^{-\|s - (i_k, j_k)\|^2}$$

*specifies that for $k \in \{1, 2, 3\}$, the $k$th subgoal is located at a single state $(i_k, j_k)$ and the artificial reward potential is a Gaussian centered at $(i_k, j_k)$. Using Figure 1 as a visual reference, one can imagine that the subgoal design $\theta = (1, 2, 8, 4, 2, 8)$ would be useful in guiding the agent toward the vicinity of the key on the right side of the room and then toward the vicinity of the goal. Once the agent is in the correct vicinity, the underlying RL algorithm can discover the precise locations of the key and goal in the particular environment realization more quickly.*

### 3.2.1 Subgoal Parameterization vs State Dimensionality

For the types of navigation tasks that we are concerned with in this paper, the dimension of the subgoal parameterization $\theta$ need not scale with the dimension of the state $s$, which would pose a potential scalability issue. Instead, one general rule-of-thumb to keep in mind is that for a dynamic subgoal exploration strategy to be effective in navigation tasks, the dimension of $\theta$ only needs to scale with the number components of $s$ that pertain to the *spatial positioning of the agent*. The next example provides an illustration.

**Example 2 (Mountain Car Environment, with $\dim(\theta) < \dim(s)$)** *Consider the well-known Mountain Car problem, a continuous control task where an underpowered car, operating in a one-dimensional space, must make its way up a steep mountain (Sutton & Barto, 2018, Example 10.1). The state is two-dimensional, $s = (x, \dot{x})$, where $x \in [-1.2, 0.5]$ is the position of the agent while $\dot{x} \in [-0.07, 0.07]$ is its velocity. A possible subgoal design with $K = 2$ is $\theta = (i_1, i_2)$, with*

$$\mathcal{G}_{\theta,k} = \{(i_k, \dot{x}) \mid \dot{x} \in [-0.07, 0.07]\} \quad and \quad \Phi_{\theta,k}(s) = e^{-(x - i_k)^2}$$

*for each $k$. In other words, the agent reaches a subgoal target state if its position is $i_k$, for any value of its velocity. Also, the artificial reward only depends on the spatial position $x$ rather than the full state $(x, \dot{x})$. In Section 5, we give numerical results for exactly this example.*

One could imagine that the concept illustrated in Example 2 also applies to more complex robotics environments with a high-dimensional state, where the number of components related to the spatial positioning is relatively small. This suggests that the subgoal parameterization (and the resulting BO problem) is often of much lower dimension than that of the state itself.

### 3.3 Subgoal-Augmented MDPs $\mathcal{M}_{\xi,\theta}$

Now that we have described how a particular subgoal design is parameterized, the remaining question is how these are integrated in a useful way into the original, sparse-reward MDP described in Section 3.1. We

propose the notion of a *subgoal-augmented*, auxiliary MDP, where the $K$ subgoals are sequentially "activated." This way, we encode subgoal ordering[5] into the exploration strategy, meaning that the agent only moves on to the next subgoal after finishing the current one.

Let $\mathcal{M}_{\xi,\theta}$ denote an auxiliary, subgoal-augmented MDP based on an original MDP $\mathcal{M}_{\xi}$, except that it is includes rewards and transitions associated with the dynamic subgoal exploration strategy $\theta$. We introduce an auxiliary state $i \in \mathcal{I} := \{0, 1, \ldots, K\}$, where $i$ represents the number of subgoals reached by the agent so far. Initially, we have $i_0 = 0$. The state of the $\mathcal{M}_{\xi,\theta}$ is $(s, i) \in \mathcal{S} \times \mathcal{I}$ and the transition for the auxiliary state is $i' = i + \mathbf{1}_{\{s' \in \mathcal{G}_{\theta,i+1}\}}$, where we take $\mathcal{G}_{\theta,K+1} = \varnothing$. This means the auxiliary state $i$ is updated to $i + 1$ whenever $s'$ reaches the next subgoal. Let the intrinsic reward of the agent be:

$$G_\theta(s_t, i_t, s_{t+1}) = \sum_{k=1}^{K} \mathbf{1}_{\{k=i_t\}} \cdot g_{\theta,k+1}(s_t, s_{t+1}),$$

where the indicator function encodes the logic that if $i_t$ subgoals have been completed so far, then the current target is subgoal $i_t + 1$ and only the rewards leading to subgoal $j + 1$ are active. The new reward function consists of both extrinsic ($R_\theta$) and intrinsic ($G_\theta$) rewards:

$$\hat{R}_{\xi,\theta}(s, i, a, s') = R_\theta(s, a, s') + G_\theta(s, i, s').$$

The value function for the new MDP $\mathcal{M}_{\xi,\theta}$ is written

$$\hat{V}_{\xi,\theta}^{\hat{\pi}}(s, i) = \mathbb{E}\left[\sum_{t=0}^{\infty} \gamma^t \hat{R}_{\xi,\theta}(s_t, i_t, a_t, s_{t+1}) \,|\, \hat{\pi}, s, i\right], \tag{3}$$

where $\hat{\pi}(\cdot|s, i)$ is now a policy defined on the new state space $\mathcal{S} \times \mathcal{I}$.

Figure 3 gives an example of the overall setup: Figure 3a shows the original MDP environment $\mathcal{M}_{\xi}$, where the dark gray cells are walls, the light gray represent uncertainty in the size of the "doors", and the red cells represent goal states (the sparse, extrinsic reward). Figure 3b shows the possible rewards the agent can encounter in the augmented MDP $\mathcal{M}_{\xi,\theta}$, for a random selection of subgoals $\theta$. The original sparse reward is represented by the red bar in the corner and the first subgoal is the one that is farther from the goal. Both subgoals are singletons and the potential functions are radial basis functions centered at the subgoal locations, similar to the parameterization described in Example 1. Note that this randomly selected set of subgoals $\theta$ is not a good exploration strategy for the environment in Figure 3a (as it leads the agent toward a wall), motivating the need for optimizing their locations, as we discuss in the next section.

## 3.4 Optimizing the Exploration Strategy

The selection of the subgoal design $\theta$ depends on the agent's underlying learning algorithm, which could in principle be any RL algorithm where intermediate rewards influence the learning process: this includes any temporal difference-based algorithm that makes use of learned value functions. In the numerical results of Section 5, our agent learns via *Q*-learning Watkins & Dayan (1992). We refer to the underlying RL algorithm as RL-ALGO. Let us use the notation RL-ALGO$[\tau, \mathcal{M}]$ to refer to the policy learned by RL-ALGO on MDP $\mathcal{M}$ after $\tau$ training interactions. We remind the reader that the subgoal-based exploration strategy is fixed *before* the test environment is revealed, so that the sequence of events in the test phase is as follows:

1. A subgoal design $\theta$ for exploration is selected.

2. The agent is placed in a new environment $\xi$.

3. The agent uses the subgoal-augmented MDP $\mathcal{M}_{\xi,\theta}$ and an RL algorithm with a budget of $\tau_{\max}$ interactions to learn a policy RL-ALGO$[\tau_{\max}, \mathcal{M}_{\xi,\theta}]$.

4. The agent's policy is evaluated using only the extrinsic reward function $R_\xi$ of the original MDP.

---

[5]Without ordering, rewards from multiple subgoals can inhibit the agent's progress.

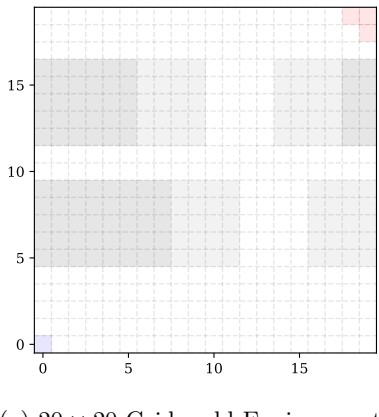
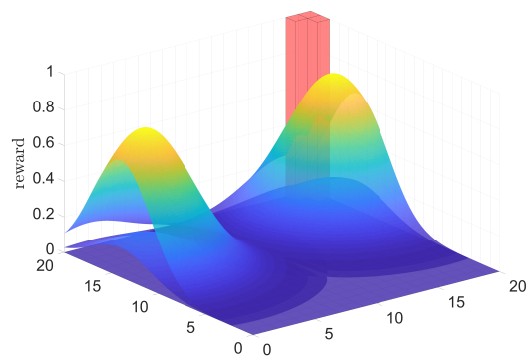

(a) $20 \times 20$ Gridworld Environment

(b) Goal and Subgoal Rewards

Figure 3: An example that visualizes an environment and a random dynamic subgoal exploration strategy along with the rewards of the associated subgoal-augmented MDP.

Our goal is to find an exploration strategy $\theta \in \Theta$ such that a policy trained using $\theta$ behaves well in the original MDP:

$$\max_{\theta \in \Theta} \ \mathbb{E}\left[\sum_{t=0}^{\infty} \gamma^t R_\xi(s_t, a_t, s_{t+1}) \,\big|\, \hat{\pi}_{\xi,\theta}^{\tau_{\max}}, (s_0, i_0)\right] \ \text{with} \ \ \hat{\pi}_{\xi,\theta}^{\tau_{\max}} = \texttt{RL-ALGO}\big[\tau_{\max}, \mathcal{M}_{\xi,\theta}\big], \tag{4}$$

where $(s_0, i_0)$ is the initial augmented state. The interpretation of the objective in (4) is as follows: Evaluate $\hat{\pi}_{\xi,\theta}^{\tau_{\max}}$ (a policy for the subgoal-augmented MDP) using the same dynamics as the subgoal-augmented MDP, but *without* the rewards associated with the subgoals. In other words, the policy takes actions based on the augmented state $(s_t, i_t)$, but only receives rewards associated with the original MDP. This explains why we need to consider a starting state $(s_0, i_0)$, rather than just $s_0$ (note that the reward does not depend on the auxiliary state $i_t$).

The expectation in (4) is taken over the random choice of a test environment $\xi$, the stochastic dynamics within $\mathcal{M}_\xi$, and the stochasticity of the learning algorithm itself. It is convenient to explicitly define the following:

$$u(\theta, \tau) = \mathbb{E}\left[\sum_{t=0}^{\infty} \gamma^t R_\xi(s_t, a_t, s_{t+1}) \,\big|\, \hat{\pi}_{\xi,\theta}^{\tau}, s_0, i_0\right] \ \text{with} \ \ \hat{\pi}_{\xi,\theta}^{\tau} = \texttt{RL-ALGO}\big[\tau, \mathcal{M}_{\xi,\theta}\big].$$

Although the objective function in (4) is $u(\theta, \tau_{\max})$, the notation $u(\theta, \tau)$ will be useful in Section 4, where we discuss using fewer than $\tau_{\max}$ interactions to learn about $u(\theta, \tau_{\max})$ as a way of reducing cost.

### 3.5 Iterative Training and Additional Cost-Reduction Levers

In our setting, we observe the performance of exploration strategies and the resulting policies in a sequence of training environment realizations $\xi^1, \xi^2, \ldots, \xi^N$ drawn from the MDP distribution $\Xi$. By default, each complete evaluation of the objective function in (4) $u(\theta, \tau_{\max})$ for a fixed $\theta$ requires running $\texttt{RL-ALGO}$ for $\tau_{\max}$ interactions. Since each interaction in the training environments is expensive (e.g., in robotics applications, this could involve time, labor, and equipment), we want to consider ways to reduce the number of training interactions. To do so, we propose two additional levers:

1. **Maximum number of interactions.** For each training environment $\xi^n$, we allow the specification of a maximum number of interactions $\tau^n$, chosen from a finite set $\mathcal{T} \subseteq \mathbb{N}$. In episodic tasks, if an episode finishes before $\tau^n$ interactions are used, we start a new episode and continue in this manner until exactly $\tau^n$ environment interactions are exhausted. In the next section, we describe our probabilistic model of the RL training curve, which allows observations of short episodes to

be informative about the final performance. This also can reduce the risk of spending too many interactions with an unpromising exploration strategy.

2. **Multiple replications.** We can reduce the variance of performance observations by averaging over the observed cumulative reward over $q^n$ i.i.d. replications, for a total of $\tau^n q^n$ interactions in training environment $\xi^{n+1}$. Each replication is an independent invocation of an agent. We suppose that $q^n$ is chosen from a finite set $\mathcal{Q}$. The idea here is that even with the same number of total interactions, a lower variance observation of a "preliminary" result could be more informative than a higher variance observation of the "full" result.

To summarize, three decisions are made at the beginning of each training opportunity $n$: (1) a choice of subgoal design $\theta^n$, (2) the maximum number of interactions $\tau^n$, and (3) the number $q^n$ of independent replications to use for this particular $\theta^n$. For each of the $q^n$ replications, we obtain a policy

$$\hat{\pi}^{\tau^n}_{\xi^{n+1},\theta^n} = \texttt{RL-ALGO}\big[\tau^n, \mathcal{M}_{\xi^{n+1},\theta^n}\big],$$

before observing a estimate of its performance. After the $q^n$ training replications are complete, we compute the average performance over the $q^n$ replications. Written more succinctly, our observation in episode $n$ takes the form

$$y^{n+1}(\theta^n, \tau^n, q^n) = u(\theta^n, \tau^n) + \varepsilon^{n+1}_{\text{env}} + \varepsilon^{n+1}_{\text{rep}}(q^n),$$

where $\varepsilon^{n+1}_{\text{env}}$ represents the deviation from the $u(\theta^n, \tau^n)$ due to the random environment $\xi^{n+1}$, while the observation noise $\varepsilon^{n+1}_{\text{rep}}(q^n)$ represents the noise that can be reduced via multiple replications, i.e., the noise in $\hat{\pi}^{\tau^n}_{\xi^{n+1},\theta^n}$ due to a sample run of $\texttt{RL-ALGO}$. Thus, $\varepsilon^{n+1}_{\text{rep}}(q^n)$ depends on the number of replications $q^n$. Naturally, a larger number of replications implies a smaller observation noise. Note that the observations $\{y^n\}$ are i.i.d., since a new MDP is sampled in each iteration. The total training cost incurred is cumulative number of interactions: $\sum_{n=0}^{N-1} \tau^n q^n$.

After training opportunities $0, 1, \ldots, N-1$, we reach the *test phase* and commit to a final subgoal design $\theta^N_{\text{rec}}$. This design is evaluated on the test MDP $\xi^{N+1} \sim \Xi$ with an agent that has a full budget of $\tau_{\max}$ interactions.

## 4 Bayesian Optimization for Cost-Efficient Exploration

The setup for BO typically consists of two components: (1) a probabilistic surrogate model (usually a Gaussian process) for modeling the objective function, and (2) an acquisition function, which given a dataset of past observations, assigns a score to each potential observation location (Garnett, 2023). Selecting the optimizer of the acquisition function as the next point to sample gives rise to strategy for balancing exploration with exploitation. The BO "loop" repeats the following: sample a point (using a combination of the current surrogate model and acquisition function), observe new data, and update the surrogate model. For a detailed tutorial, we refer readers to Frazier (2018) and Garnett (2023).

For our setting of learning a dynamic subgoal exploration strategy, we propose a tailored probabilistic model for the RL learning curve and an acquisition function for selecting the next subgoal design, the maximum episode length, and the number of replications to run. Although shorter episodes and smaller number of replications are more cost-efficient, they also decrease the chance of reaching the goal and produce higher observation noise. Thus, the acquisition function must carefully trade off these downsides with the cost of interactions. We call this the *Bayesian Exploratory Subgoal Design* (BESD) acquisition function.

### 4.1 Surrogate Model

In order to enable the ability to dynamically select the maximum episode length of training, as described in Section 3.5, our approach uses a GP surrogate model over $u(\theta, \tau)$, rather than $u(\theta, \tau_{\max})$. In other words, our model is a function of both $\theta$ and $\tau$ rather than just $\theta$, enabling it to capture the performance of a policy trained with subgoals $\theta$, for a variety of episode lengths. Assume that $\Theta \subseteq \mathbb{R}^m$. We place a GP prior $f$ on the latent function $u$ with mean function $\mu : \Theta \times \mathcal{T} \to \mathbb{R}$ and covariance function $k : (\Theta \times \mathcal{T}) \times (\Theta \times \mathcal{T}) \to \mathbb{R}_+$. More precisely, to capture the structure of the RL learning curve, we set $\mu$ to the mean of an initial set

---

**Algorithm 1** Bayesian Exploratory Subgoal Design

---

1. Set $n = 0$. Estimate hyperparameters of the GP prior $f$ using initial samples.

2. Compute next decision $(\theta^n, \tau^n, q^n)$ according to the acquisition function (7).

3. Train in environment $\xi^{n+1}$ augmented with $\theta^n$ $(\mathcal{M}_{\xi^{n+1}, \theta^n})$ using levers $(\tau^n, q^n)$.

4. Observe $y^{n+1}(\theta^n, \tau^n)$ and update posterior on $f$.

5. If $n < N$, increment $n$ and return to Step 2.

6. Return a subgoal recommendation $\theta^N_{\text{rec}}$ that maximizes $\mu^N(\theta, \tau_{\max})$.

---

of samples and use a multidimensional product kernel, based on the kernel used in Klein et al. (2017) for modeling ML performance as a function of parameters and time:

$$k\big((\theta, \tau), (\theta', \tau')\big) = k_\theta(\theta, \theta') \, k_\tau(\tau, \tau'), \tag{5}$$

where the first kernel $k_\theta$ is the (5/2)-Matérn kernel and $k_\tau$ is a polynomial kernel $k_\tau(\tau, \tau') = \phi(\tau)^\intercal \Sigma_\phi \, \phi(\tau')$ with $\phi(\tau) = (1, \tau)^\intercal$ and hyperparameters $\Sigma_\phi$. Note that the covariance under $k$ is large only if the covariance is large under both $k_\theta$ and $k_\tau$. We make the modeling assumption that $\varepsilon^{n+1}_{\text{env}}$ and $\varepsilon^{n+1}_{\text{rep}}(q^n)$ are independent, zero mean, and normally distributed[6] with variances $\sigma^2_{\text{env}}$ and $\sigma^2_{\text{rep}}/q^n$, respectively. This allows us to take advantage of standard GP machinery to analytically compute the posterior on $f$ conditioned on the history after $n$ steps. This posterior is another GP, whose mean and kernel functions are denoted $\mu^n(\theta, \tau)$ and $k^n((\theta, \tau), (\theta', \tau'))$; the exact expressions can be found in, e.g., Rasmussen & Williams (2006).

We remind the reader that the dimensionality of the GP surrogate model is $\dim(\Theta) + 1$, i.e., the dimension of the subgoal parameterization, along with an additional dimension for $\tau$. As illustrated in Example 2, it will often be the case for navigation domains that the dimension of the subgoal parameterization is smaller than that of the state space of the underlying RL problem (due to the relatively small number of spatial components of the state). Therefore, dynamic subgoal exploration strategies can be tractably modeled and optimized for broad classes of navigation problems, even with vanilla GPs.[7]

### 4.2 Acquisition Function

As described above, the proposed algorithm proceeds in iterations, selecting one set of subgoals $\theta^n$ along with $\tau^n$ and $q^n$, to be evaluated in each training environment. We now propose the acquisition function for making these evaluation decisions. An overview of the BO setup is given in Algorithm 1.

Suppose the training budget is used up after training iterations $0, 1, \ldots, N-1$. Then, the optimal risk-neutral decision is to use subgoals on the test MDP $\xi^{N+1}$ that have maximum expected performance under the posterior. The expected score of this choice is $\mu^n_*$ where

$$\mu^n_* := \max_\theta \mu^n(\theta, \tau_{\max}), \tag{6}$$

where $\mu^n(\theta, \tau_{\max}) = \mathbb{E}_n[f(\theta, \tau_{\max})]$. Here $\mathbb{E}_n$ is the conditional expectation with respect to the history after the first $n$ observations: $(\theta^0, \tau^0, q^0, y^1, \ldots, \theta^{n-1}, \tau^{n-1}, q^{n-1}, y^n)$. Note that although we are allowed to use fewer than $\tau_{\max}$ interactions in training environments to reduce cost, the agent uses its full budget for the test MDP $\xi^{N+1}$.

The proposed acquisition function is based upon the *knowledge gradient*, which is one-step lookahead approach (Frazier et al., 2008; 2009). That means we imagine for each training MDP that it is the last opportunity before the test MDP and act optimally. Full lookahead approaches require solving an intractable dynamic

---

[6]Although the assumption of normality is commonplace in BO for tractability of the posterior (Frazier, 2018), other noise distributions can be used through an appropriate likelihood function.

[7]When the need arises to optimize for high dimensional subgoal parameterizations, one may opt for scalable extensions of the model and optimization formulation; see, e.g., Wang et al. (2016); Mutny & Krause (2018); Nayebi et al. (2019); Eriksson et al. (2019); Papenmeier et al. (2022; 2023). We leave extensions in this direction to future work and focus on a more standard setting.

programming problem; however, we show that nonetheless, the one-step approach is asymptotically optimal in Theorem 1 and Theorem 2. If we evaluate $(\theta, \tau, q)$, i.e., the subgoals $\theta$ for $\tau$ steps and $q$ replications, then the expected gain in performance in the test MDP of the recommended exploration strategy after the evaluation, based on (6), with respect to the current best is

$$\nu^n(\theta, \tau, q) = \mathbb{E}_n\left[\mu_*^{n+1} \mid \theta^n = \theta, \tau^n = \tau, q^n = q\right] - \mu_*^n.$$

Therefore, the one-step optimal strategy is to choose the next subgoals $\theta^n$, maximum episode length $\tau^n$, and number of replications $q^n$ so that $\nu^n$ is maximized. However, this strategy would generally allocate a maximum number of steps $\tau_{\max}$ and replications $q_{\max}$ for the evaluation of the next subgoal design, as observing $\tau_{\max}$ during training is most informative, and repeating for $q_{\max}$ replications reduces the noise maximally. In other words, this strategy does not consider the cost of training.

Hence, we propose an acquisition strategy that maximizes the *gain in performance per effort*:

$$(\theta^n, \tau^n, q^n) \in \underset{\theta, \tau, q}{\arg\max} \ \frac{\nu^n(\theta, \tau, q)}{q\tau}. \tag{7}$$

The optimization problem (7) is challenging when the domain $\Theta$ is continuous, so we take the approach of replacing it with a discrete domain $\bar{\Theta} \subseteq \Theta$ (for example, this could be selected by a Latin hypercube design (Stein, 1987)). This approach has been applied successfully in other knowledge gradient style acquisition functions (Scott et al., 2011; Wu & Frazier, 2016; Poloczek et al., 2017). We provide a novel theoretical guarantee on the asymptotic suboptimality of a discretized optimization domain. It characterizes the Lipschitz constant explicitly, thereby improving on the analysis of Poloczek et al. (2017); see Theorem 2 in the next section.

### 4.3 Theoretical Analysis

We now provide our main theoretical results on the asymptotic optimality of BESD. Detailed proofs can be found in Appendix A. For convenience, we suppose $\mu(\theta, \tau) = 0$ for all $(\theta, \tau)$, and further assume that the kernel $k(\cdot, \cdot)$ has continuous partial derivatives up to the fourth order. Recall that $\theta_{\text{rec}}^N \in \bar{\Theta}$ is the final recommendation made in iteration $N$:

$$\theta_{\text{rec}}^N \in \underset{\theta \in \bar{\Theta}}{\arg\max} \ \mu^N(\theta, \tau_{\max}).$$

Our first theorem is concerned with the finite, discretized optimization domain $\bar{\Theta}$.

**Theorem 1** *The acquisition function described in (7) has the property of asymptotic optimality with respect to $\bar{\Theta}$, i.e.,*

$$\lim_{N \to \infty} f(\theta_{rec}^N, \tau_{max}) = \max_{\theta \in \bar{\Theta}} f(\theta, \tau_{max}),$$

*almost surely. That is, the recommended design $\theta_{rec}^N$ becomes optimal as $N \to \infty$.*

If the optimization domain $\bar{\Theta} = \Theta$, then Theorem 1 suffices. Unfortunately, for many applications, the subgoal parameterizations will naturally be continuous. Next, we provide an additive bound on the difference between the solution of BESD in $\bar{\Theta}$ and the unknown optimum in $\Theta$, as the number of iterations $N$ tends to infinity.

We use a probabilistic Lipschitz constant of a GP from Lederer et al. (2019) to quantify the performance with respect to the full, continuous subgoal parameter space. We make use of the fact that the derivative $df(\theta, \tau_{\max})/d\theta_i$ is another GP with covariance

$$k^{\partial i}(\theta, \theta') = \frac{\partial^2}{\partial \theta_i \partial \theta_i'} \ k\big((\theta, \tau_{\max}), (\theta', \tau_{\max})\big),$$

for all $i = 1, 2, \ldots, m$ (Rasmussen & Williams, 2006, Section 9.4). See also Ghosal et al. (2006) and Wu et al. (2017) for other uses of this property. For each $i = 1, 2, \ldots, m$, define the constant

$$L_\delta^i = k_{\max}^\partial \sqrt{2 \log\left(\frac{2m}{\delta}\right)} + 12\sqrt{6m} \max\left\{k_{\max}^\partial, \sqrt{L_k^{\partial i} \max_{\theta, \theta' \in \Theta} \text{dist}(\theta, \theta')}\right\}, \tag{8}$$

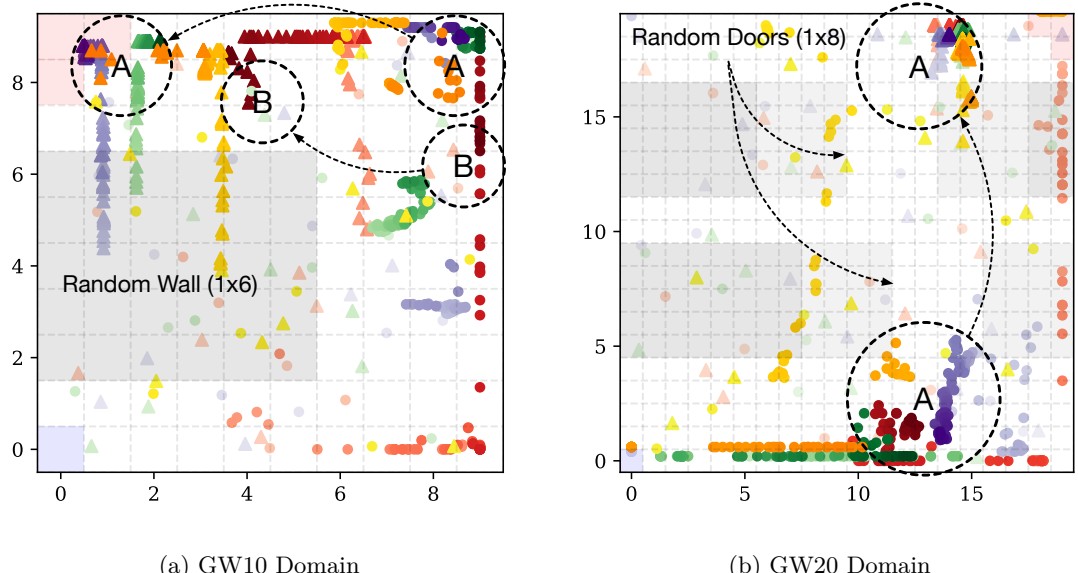

(a) GW10 Domain

(b) GW20 Domain

Figure 4: Recommendation paths for GW10 and GW20. The blue and red shaded regions denote the starting points and goals, respectively. Dark and light gray regions possible locations of walls and doors, respectively. Each plot displays four realizations of the "recommendation paths" of BESD. Each color corresponds to one sample realization, and the color becomes darker as $n$ increases, with the lightest points being the initial samples. The circles and triangles represent the first and second subgoals, respectively, of the exploration strategy. The 'A' and 'B' labels point out two example sets of subgoals displaying notable behaviors.

where $L_k^{\partial i}$ be a Lipschitz constant of the kernel $k^{\partial i}$ and $k_{\max}^{\partial} = \max_{\theta \in \Theta} \sqrt{k^{\partial i}(\theta, \theta)}$.

**Theorem 2** *The acquisition function of (7) has bounded asymptotic suboptimality with respect to the original domain $\Theta$ in the sense that with probability at least $1 - \delta$, it holds that*

$$\lim_{N \to \infty} f(\theta_{\mathrm{rec}}^N, \tau_{\max}) \geq \max_{\theta \in \Theta} f(\theta, \tau_{\max}) - d \cdot \|L_\delta\|$$

*where $d = \max_{\theta \in \Theta} \min_{\theta' \in \bar{\Theta}} \mathrm{dist}(\theta, \theta')$ is a measure on the "coarseness" of the discretization and $L_\delta$ is the vector $(L_\delta^1, L_\delta^2, \ldots, L_\delta^m)$, with each $L_\delta^i$ defined as in (8).*

## 5 Numerical Experiments

We now show numerical experiments to demonstrate the cost-effectiveness of the BESD framework. BESD is implemented using the MOE package (Clark et al., 2014) and the full source code be found at the following URL: https://github.com/yjwang0618/subgoal-based-exploration. In the experiments that follow, we use the BESD approach to optimize dynamic subgoal exploration strategies consisting of two or three subgoals.

BESD is given a few choices for the interaction length $\tau$ and number of replications $q$ (values reported for each benchmark below). Each replication of the BESD is given an initial set of 10 observations for each value of $\tau$ (these initial observations incur interaction costs just like future observations). The potential function at state $s$ with the $j$th subgoal activated is $\Phi_j(s) = w_1 \exp[-0.5(s - j)^2 / w_2]$, where the "height" is $w_1 = 0.2$ and "width" is $w_2 = 10$. The underlying RL algorithm for all environments is Q-learning with an $\epsilon$-greedy behavioral policy (with $\epsilon = 0.2$) for all environments.

In the next few subsections, we first introduce each of the environments and along the way, show some qualitative results obtained by BESD, focusing on providing intuition. Later, in Section 5.6, we introduce the baseline algorithms used for an empirical comparison, and in Section 5.7, provide a discussion of those results.

### 5.1 Windy Gridworlds with Walls

The first set of environments (GW10) is a distribution over $10 \times 10$ gridworlds, where the goal is to reach the upper left square that is shaded red in Figure 4a to collect a reward of one. The agent starts from the lower-left grid square shaded in blue and may in each step choose an action from the action space consisting of the four compass directions. Each gridworld is partitioned by a wall into two rooms. The wall, randomly located in one of the middle five rows in the gridworld, has a door located on four grid squares on its right. The agent will stay in the current location when it hits the wall.

There is a small amount of "wind" or noise in the transition: the agent moves in a random direction with a probability that is itself uniformly distributed between 0 and 0.02 (thus, a particular environment instance drawn from the distribution has a random wall location and wind probability).

We use $\mathcal{T} = \{200, 600, 1000\}$ for the possible values of $\tau$ and $\mathcal{Q} = \{5, 20\}$ for the possible values of $q$. We parameterize the exploration strategy using two subgoals, whose locations are optimized. Subgoal locations are limited to the continuous subset of $\mathbb{R}^2$ which contains the grid, i.e., $\Theta = ([0, 10] \times [0, 10])^2$ for GW10.

#### 5.1.1 Recommendation Paths for GW10

In order to visualize the qualitative behavior of `BESD`, we show in Figure 4a the evolution of the recommended subgoals over time (iterations), a concept that we refer to as a *recommendation path*. The plot displays four recommendation path realizations of `BESD` using distinct colors. Within each color, the lightest points are the initial samples while the darker points represent recommendations for larger $n$. Also within each color, the circles represent the first subgoal of the exploration strategy, while the triangles represent the second subgoal. We point out two types of exploration behaviors discovered by `BESD` in Figure 4a:

- Behavior 'A': The pairs of regions labeled 'A' are the final recommendations of the orange, green, and purple sample paths. The strategy leads the agent toward the upper right corner (away from the wall), and then after that, directly towards the goal.

- Behavior 'B': The final recommendation of the red sample path is labeled by 'B.' Note that in behavior 'A', a direct path to the first subgoal (upper right corner) is blocked by the random wall for some realizations of the environment. Behavior 'B' might be interpreted as a slight remedy of this situation by targeting a lower region of the right edge, creating a more direct path around the wall.

Both strategies appear to be reasonable ways for the agent to avoid the door and head to the goal.

### 5.2 Larger, Three-Room Windy Gridworlds

The second domain (GW20) is a distribution of larger $20 \times 20$ gridworlds with three rooms separated by two walls. As shown in Figure 4b, the walls are randomly located in the middle rows (dark gray). A door of size 8 is randomly located somewhere within the wall, shaded in light gray. The starting location is the blue square in the lower left and the goal is displayed in red in the upper right. As in GW10, we optimize the locations of a two-subgoal exploration strategy, with $\Theta = ([0, 20] \times [0, 20])^2$. The noise due to wind is the same as in GW10. In this experiment, we consider the case of only allowing `BESD` to select the maximum episode length from $\mathcal{T} = \{4000, 7000, 10000\}$, while keeping $q = 20$ fixed.

#### 5.2.1 Recommendation Paths for GW20

Recommendation paths are shown in Figure 4b. Unlike the case of GW10, all four of the realizations converge to roughly the same exploration strategy, labeled by 'A.' Focusing on the lighter red and orange circles, we can notice a trend of the first subgoal initially being placed (naively) near the goal, but as learning progresses, they move downward toward the entrance of the first door. The second subgoal converges toward the exit of the second door, moving the agent near the goal.

Regarding the placement of the first subgoal near the goal and inducing a direct path, it is worth pointing out this strategy might work for *some* environments (i.e., those where the first door is at its leftmost position

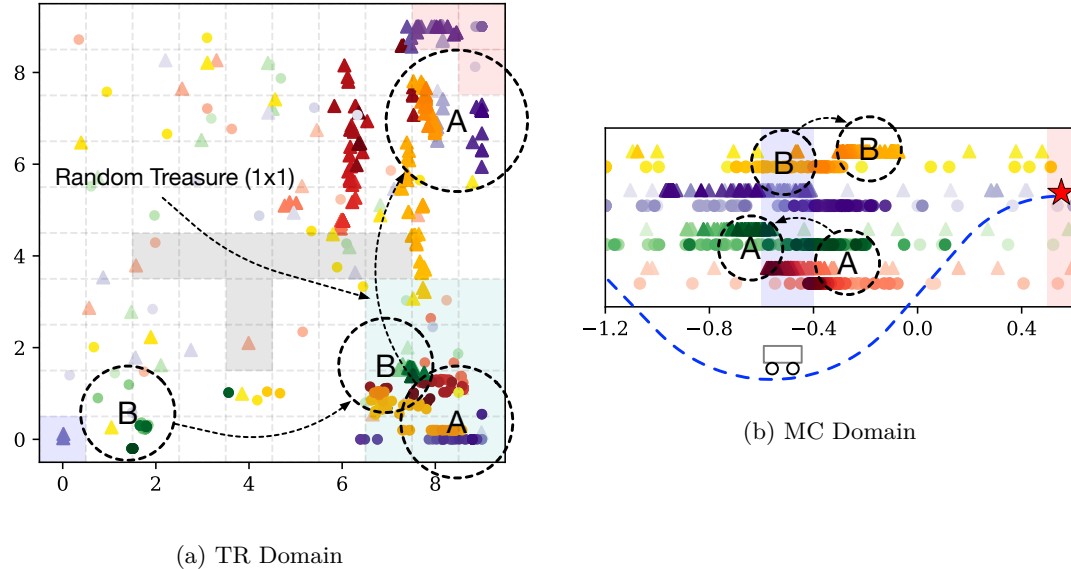

(a) TR Domain

(b) MC Domain

Figure 5: Recommendation paths for TR and MC. The first panel, Figure 5a, largely follows the same design as Figures 4a and 4b, except that the green squares represent possible location of the treasure. In the second panel, Figure 5b, since the location of the mountain-car is one-dimensional, we visualize the four recommendation paths by spacing them vertically to avoid crowding (therefore, the vertical axis represents different trajectories, each shown in a different color). The initial location of the car is colored in blue, while the goal is in red, corresponding to the overlay of the mountain. See the caption of Figure 4 for an explanation of the symbols used.

and the second door is at its rightmost position). However, BESD learns that in order to perform well *across the distribution* of environments, the strategy of first moving rightward is better.

## 5.3 Treasure-in-Room

The third domain (TR) is a distribution of $10 \times 10$ gridworlds with a "treasure" hidden in a small room; see Figure 5a. The light green area shows the possible positions of the treasure. The agent gets a reward of 10 upon entering the square with treasure, and a reward of 10 upon reaching the goal. The cumulative reward, however, is zero if the agent does not find the goal within the interaction budget. The discount factor is set to $\gamma = 0.98$ to encourage policies that collect the reward earlier. We set $\mathcal{T} = \{400, 1200, 2000\}$ and $\mathcal{Q} = \{5, 20\}$.

### 5.3.1 Recommendation Paths for TR

The recommendation paths for TR are shown in Figure 5a. We observe that two strategies were discovered by BESD across these four realizations:

- Behavior 'A': This appears to be the ideal behavior and was discovered in the orange, purple, and red sample paths: first lead the agent to the treasure and then toward the goal through the upper right. It is also notable that the first subgoal is located at the *bottom* of the room, meaning that wherever the treasure turns out to be, the agent can pick it up without backtracking.

- Behavior 'B': The green sample path's final recommendation coincides with the (apparently suboptimal) exploration strategy denoted by 'B' simply leads the agent to the treasure, but does not provide any guidance toward the goal. We highlight that this is an instance where BESD's learning is not yet complete, evidenced by the fact that behavior 'B' is often recommended in *earlier iterations of the orange sample path*. In that case however, BESD eventually discovers behavior 'A' in later iterations.

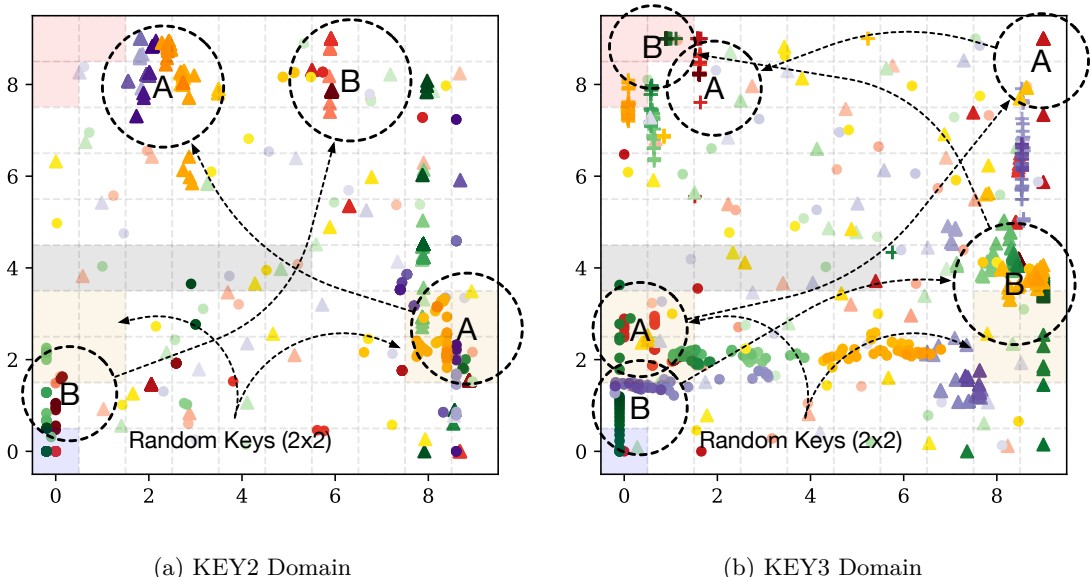

(a) KEY2 Domain        (b) KEY3 Domain

Figure 6: Recommendation paths. The blue and red shaded regions denote the starting points and goals, respectively. Dark and light gray regions possible locations of walls and doors, respectively. Each plot displays four realizations of the "recommendation paths" of BESD. Each color corresponds to one sample realization, and the color becomes darker as $n$ increases, with the lightest points being the initial samples. The circles, triangles, and crosses represent the first, second, and third subgoals, respectively. The 'A' and 'B' labels point out two example sets of subgoals displaying notable behaviors.

## 5.4 The Mountain Car Problem (MC)

The mountain car (MC) domain, as we introduced in Example 2, is a commonly used RL benchmark environment that tests an agent's ability to explore, as it is required to go in the opposite direction of the goal in order to reach the top of the mountain; see, e.g., (Sutton & Barto, 2018, Example 10.1). For this experiment, we created a distribution of environments $\Xi$ by randomizing the starting location of the agent, which is chosen uniformly from $[-0.6, -0.4]$. Here, we set $\mathcal{T} = \{4000, 7000, 10000\}$ and $\mathcal{Q} = \{10, 50\}$.

### 5.4.1 Recommendation Paths for MC

The subgoal-pairs discovered by BESD are shown in Figure 5b; they tend to be on opposite sides of the agent's starting location, thereby creating back-and-forth movement needed to generate momentum and move up the mountain. It is worth noting that the symmetric behaviors of going from left to right (Behavior 'B' in Figure 5b, for the orange sample path) and going from right to left (Behavior 'A', exhibited by the green, red, and purple sample paths) can both be found in the results of BESD.

## 5.5 Key-Door with Highly Varying Key Locations (KEY2 and KEY3)

In our last experiment, we test for the situation where the distribution of environments $\Xi$ contains environments that might vary dramatically from one another. We also consider how the exploration behavior changes when we add an additional subgoal to the strategy.

In domains KEY2 (with two subgoals) and KEY3 (with three subgoals), we consider a $10 \times 10$ gridworld with one wall, where a "key" needs to be picked up before opening a closed door at the upper-right corner of the grid. The location of the key, however, is highly varying and is either near the left wall or the right wall. The environment is visualized in Figures 6a and 6b. We set $\mathcal{T} = \{400, 700, 1000\}$ and $\mathcal{Q} = \{5, 20\}$.

### 5.5.1 Recommendation Paths for KEY2/KEY3

It is important that the agent moves in the *vicinity of both keys* in order for it to perform well across the distribution of environments. We now discuss how this is achieved by the two- and three-subgoal exploration strategies, using the annotations in Figures 6a and 6b.

- Behavior 'A' in KEY2 (Figure 6a): In the first exploration behavior discovered by BESD, the agent is first directed to the right-most key location and then towards the door. This is behavior is reasonable in the sense that the agent's initial location is near the left-most key location; hence, the naive exploration (e.g., $\epsilon$-greedy) "built-in" to RL-ALGO would likely find the key (if it is there) without additional subgoal rewards.

- Behavior 'B' in KEY2 (Figure 6a): The second exploration behavior that we highlight takes a similar approach. This strategy incentivizes the agent to first check the left-most key location (going upwards from the initial location). Interestingly, the second subgoal is neither the other key location nor the goal: instead, the agent is directed toward the upper edge of the environment, slightly right of center. Upon examination, one might conclude that this path *compromises* between the second key location and the goal. On its way from the first to second subgoal, the agent enters the vicinity of the second key location and also ends up not far from the goal. In other words, the exploration strategy puts the agent in a position such that RL-ALGO's naive exploration is more likely to be successful.

- Behavior 'A' in KEY3 (Figure 6b): With an additional subgoal to work with, BESD is able to find more flexible exploration strategies. For behavior 'A', we see that the first subgoal is near the left-most key location, the second subgoal indirectly leads the agent toward the vicinity of the right-most key location, and the third subgoal is at the goal. The placement of the second subgoal is reminiscent of behavior 'B' of KEY2, but this time, a third subgoal allows BESD to directly lead the agent towards the goal

- Behavior 'B' in KEY3 (Figure 6b): This strategy is more intuitive (indeed, more replications converge to behavior 'B' than behavior 'A') and leads the agent to check each of the possible key locations (the closer one first) and then sends the agent directly toward the goal.

### 5.6 Baseline Algorithms

Given the somewhat unique positioning of the BESD framework, it is important for us to compare against from several streams of literature. Due to our strong focus on cost-efficiency, non-gradient-based approaches from the BO literature are particularly relevant. Two of the most common approaches are *expected improvement* (Močkus, 1975; Jones et al., 1998) and *lower confidence bound* (LCB)(Cox & John, 1992; Srinivas et al., 2010). Expected improvement (EI) allocates one sample in each round, selecting a point that maximizes the expected improvement beyond currently sampled points:

$$\texttt{EI}(\theta) = \mathbb{E}_n\left[\left(\min\{y^1, \ldots, y^n\} - y^{n+1}(\theta, \tau_{\max})\right)^+\right].$$

In each iteration, we evaluate the EI selection using $\tau_{\max}$ iterations. LCB controls the exploration-exploitation trade-off using a "bonus term" proportional to the standard deviation at each point:

$$\texttt{LCB}(\theta) = \mu^n(\theta, \tau_{\max}) - \kappa\sqrt{k^n((\theta, \tau_{\max}), (\theta, \tau_{\max}))}.$$

The parameter $\kappa$ is set to 2. Both EI and LCB are implemented using the GPyOpt package González (2016). As a sanity check, we also compare against a baseline where the subgoals are randomly selected at each iteration (RND), implemented using Latin hypercube sampling (Stein, 1987).

We also compare against two "default RL" baselines, that do not incorporate an aspect of tuning the exploration strategy. The first baseline is the $Q$-learning algorithm (QL) (Watkins, 1989) with no subgoals or reward shaping: that is, we directly run QL on environment $\xi^N$ for $\tau_{\max}$ interactions. The second one is a heuristic based on the approximate Q-values learned by QL, which we call "transfer" $Q$-learning (TQL): for the test instance, we initialize the $Q$-values using the previously stored results from a randomly chosen training

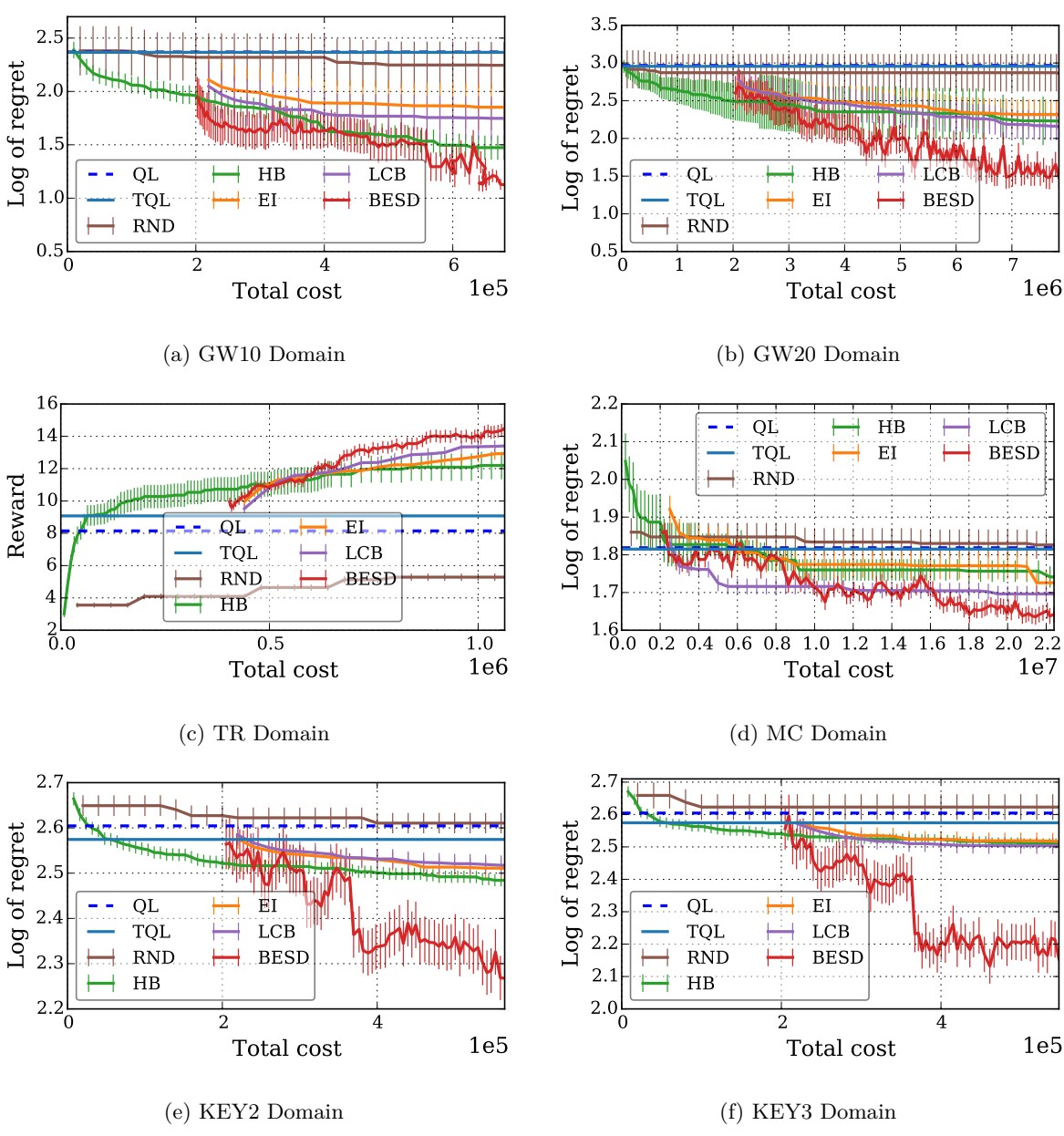

Figure 7: Performance as a function of the total training costs. The curves are averaged over 50 replications of the meta-optimization problem and the error bars indicate $\pm$ 2 standard errors of the mean. Note that the curves associated with the BO methods, BESD, LCB, EI, start later due to the use of a set of initial points for initializing the GP model.

environment. This heuristic is inspired by the idea of *policy reuse* proposed in Fernández et al. (2010) for transferring learned strategies to new tasks.

An alternative to applying BO or bandit algorithms to hyperparameter optimization is the idea of *adaptive configuration evaluation*, which focuses on improving the throughput of configuration evaluation by quickly eliminating ones that are not promising. From this line of thinking, the Hyperband algorithm (HB) of Li et al. (2017) stands out as a popular and representative approach. It treats hyperparameter optimization as a pure-exploration infinite-armed bandit problem; it uses sophisticated techniques for adaptive resource allocation and early-stopping to concentrate its learning efforts on promising designs. Setting $\eta = 3$ (the

default value) and $R = 81$, HB consists of $\lfloor \log_\eta R \rfloor$ rounds. The first round starts with $R$ samples of subgoal designs $\theta$ from a Latin hypercube sample. Following HB's motivation of early-stopping unpromising designs, each $\theta$ is evaluated for $\tau_{\min}$ steps. The best $1/\eta$-fraction designs are kept for the next round. In round $i$, Hyperband samples $R/\eta^{i-1}$ subgoal designs to evaluate for $\tau_{\min} \eta^{i-1}$ steps.

A detailed empirical comparison of BESD to baseline algorithms for all environments are given in Figure 7. The purpose of each numerical experiment is to show that BESD is able to, in a cost-efficient manner (where cost is defined as the number of environment interactions), produce exploration strategies that lead to policies that perform well in a randomly drawn test environment. For each replication, to assess the performance at a particular point in the process, we take its latest recommendation and test it by averaging its performance on a random sample of 200 test MDPs (i.e., $\xi^N$). The $x$-axis is the cumulative cost, which includes the initial sampling cost. The $y$-axis is typically the log regret (lower is better), where regret is defined as the number of additional steps needed to reach the goal when compared to the optimal policy. The exception is the TR domain, where the $y$-axis is the discounted reward (higher is better), since in TR, the performance is measured by both reward and steps.

### 5.7 Takeaways from Baseline Comparisons in Figure 7

We now offer some observations and takeaways from the performance plots of Figures 7a-7f, where BESD is compared to a variety of baseline approaches. In these figures, the $x$-axis shows the total cost, defined to be the total number of environment interactions, and the $y$-axis shows the performance measure (either regret or reward, depending on the environment) when averaged across environments drawn from $\Xi$. In all environments, BESD (in red) achieves either the lowest regret or highest reward.

1. **Sanity checks.** The methods RND, QL, and TQL tend to perform poorly across all domains. This suggests that subgoal-free methods (QL and TQL) are unable to achieve cost-efficient exploration. At the same time, the results of RND suggest that when using subgoals, they must be carefully selected (i.e., exploration based on random subgoals does not perform well).

2. **Comparison to Hyperband.** HB is reasonably competitive against BESD on two of the easier domains, GW10 and TR. In particular, we notice that HB tends to have good performance early on (as it is able to use early stopping to quickly eliminate inferior subgoal strategies). However, as the interaction budget grows, we see that in most domains, BESD is eventually able to make better use of its evaluations, likely explained by BESD's use of a tailored surrogate model.

3. **Comparison to other BO methods.** The popular BO methods EI and LCB tend to perform similarly to each other in all domains. Compared to BESD, however, they are less cost-efficient. Since all three approaches make use of underlying GP surrogate models, but EI and LCB are constrained in always using $q_{\max}\tau_{\max}$ interactions, this is evidence that being able to reduce the episode lengths and the number of replications is valuable.

4. **Impact of more subgoals.** Lastly, we point out that Figures 7e and 7f show that although a two-subgoal exploration strategy achieves better results than the baselines, a three-subgoal strategy performs even better. This demonstrates the benefit of expanding the dimension of the parameterization in certain environments. Choosing the number of subgoals to use in a particular set of environments is not an exact science; in general, a higher dimensional subgoal parameterization makes the BO meta-optimization problem more challenging and each acquisition function optimization is also more time-consuming. We recommend the following guidelines: (1) Consider the total interaction budget across all training iterations. A rule-of-thumb is that a $d$-dimensional subgoal parameterization should have $2d - 1$ random initial points. The interaction cost of the initial points should be less than $1/3$ of the total budget in order to give BESD adequate time to make progress (if the cost of initial points is too high, then one might want to reduce $d$). (2) Optimizing the acquisition function becomes more time consuming as $d$ increases, so $d$ should be small enough such that (7) can be computed in one's allotted per-iteration time budget for acquisition function optimization.

### 5.8 Dynamic Subgoal Exploration Strategy vs. Learning From Scratch at Test Time

In Section 5, Figures 4, 5, and 6 gave visual intuition about the types of exploration behaviors that were discovered by BESD. In this section, we show how the final dynamic subgoal strategy $\theta_{\text{rec}}^N$ recommended by BESD is able to speed up learning in the test environment, by comparing it to "learning from scratch" (i.e., running RL directly in the sparse reward environment, with no subgoals). Let

$$\pi_\xi^T = \texttt{RL-ALGO}\big[T, \mathcal{M}_\xi\big] \quad \text{and} \quad \pi_{\xi,\,\theta_{\text{rec}}^N}^T = \texttt{RL-ALGO}\big[T, \mathcal{M}_{\xi,\,\theta_{\text{rec}}^N}\big]$$

be the policy learned using RL-ALGO on the original, sparse-reward environment (i.e., no subgoals) and the policy learned by RL-ALGO with the aid of the subgoal strategy found by BESD after $T$ test-time interactions. The performance ratio that we are interested in is

$$\text{performance ratio}(T) = \mathbb{E}\Big[V^{\pi_{\xi,\,\theta_{\text{rec}}^N}^T}(s_0)\Big] \,/\, \mathbb{E}\big[V^{\pi_\xi^T}(s_0)\big],$$

which, stated simply, represents the ratio "performance with subgoals / performance without subgoals." On GW10, GW20, MC, KEY2, and KEY3, a smaller performance ratio indicates a more effective exploration strategy. For TR, we measure performance using rewards instead of costs, so a larger performance ratio is better. Table 1 displays the performance ratios as a function of the number of interactions used in the test environment. We can see that an optimized exploration strategy corresponds to dramatic improvements, ranging from roughly 3x in the worst cases (MC, KEY2, and KEY3) to nearly 20x in the best cases (GW10, GW20, and TR). Note that due to the varying difficulty between environments, we use a scaling factor $m$ to show how the performance improves with additional cost. The takeaway here is that using a good exploration strategy can lead to dramatic improvements at test-time.

Table 1: Performance ratios as a function of interactions in the test environment. GW10, GW20 TR, MC, KEY2, and KEY3 are evaluated every $m = 100, 1000, 200, 1000, 500, 500$ steps respectively.

| $\tau$ | GW10 | GW20 | TR | MC | KEY2 | KEY3 |
|---|---|---|---|---|---|---|
| $m$ | 0.458 | 0.779 | 0.436 | 0.980 | 1.456 | 1.025 |
| $2m$ | 0.218 | 0.492 | 2.823 | 1.048 | 0.736 | 0.940 |
| $3m$ | 0.086 | 0.234 | 2.823 | 0.949 | 1.277 | 0.698 |
| $4m$ | 0.080 | 0.224 | 0.917 | 0.896 | 0.704 | 0.788 |
| $5m$ | 0.070 | 0.108 | 6.723 | 0.987 | 1.355 | 0.531 |
| $6m$ | 0.086 | 0.088 | 8.939 | 0.878 | 0.856 | 0.503 |
| $7m$ | 0.080 | 0.068 | 9.908 | 1.077 | 0.920 | 0.623 |
| $8m$ | 0.087 | 0.075 | 10.216 | 0.877 | 0.883 | 0.532 |
| $9m$ | 0.069 | 0.059 | 23.2936 | 0.512 | 0.232 | 0.566 |
| $10m$ | 0.069 | 0.058 | 18.011 | 0.354 | 0.332 | 0.361 |

## 6 Conclusion and Future Work

The problem of finding exploration strategies for a distribution of environments with a strong focus on *cost-awareness during training* has not been adequately studied in the literature. This can be a deterrent to applying RL in real-world settings where interactions with the environment are limited and expensive (and where cheap simulators are not available). This paper proposes a solution based on Bayesian optimization; in a cost-aware manner, our approach finds subgoals with an intrinsic shaped reward that aids the agent in scenarios with sparse and delayed rewards, thereby reducing the number of interactions needed to obtain a good solution. We hope that this approach can help RL become more applicable in real world settings. An experimental evaluation demonstrates that BESD achieves considerably better solutions than a comprehensive field of baseline methods on a variety of benchmark problems. Moreover, an examination of its "recommendation paths" shows that BESD discovers solutions that induce interesting exploration strategies. There are several exciting directions for extending this paper:

- **Richer BO formulations.** Extensions to the BO formulation could be made in various ways. For example, one interesting direction is to allow the acquisition function to determine the *number of subgoals* as an additional lever. Based on a few informal observations, such a formulation is likely only interesting in settings where *more subgoals incur additional experimentation cost.*[8] Alternatively, the acquisition function itself could be extended with additional features, such as encouraging successive subgoal evaluations to be nearby previous ones (i.e., to reduce setup cost) or the ability to reason about (known) symmetries in the domain. Such advanced features might be enabled by dynamic programming formulations *of the BO problem itself*, which can be tackled using multi-step lookahead BO (Lam et al., 2016; González et al., 2016; Jiang et al., 2020; Lee et al., 2020). Other possiblities include the ability to handle expensive-to-evaluate constraints (Gardner et al., 2014; Gelbart et al., 2014; Letham et al., 2019) or total cost budgets (Astudillo et al., 2021; Lee et al., 2021).

- **Case study in an application domain.** Our experiments gave proof-of-concept results on benchmarks where the RL training itself did not use prohibitive amounts of computation, in order for us to stay within a reasonable computational budget. This is because statistically distinguishable results for baseline algorithms require many replications of the meta-optimization problem (i.e., the BO routines), each of which require many iterations of RL training. One immediate area of future work is to "productionize" the dynamic subgoal exploration strategies in a real-world application involving a navigation task.

- **The task-aware setting.** Finally, our problem formulation does not include "labels" for environments, as our setting is concerned with case of exogenous variation in the environments, but otherwise the same task. The situation often studied in the multi-task RL setting, however, often comes with task identifiers, where the agent knows that it is operating in particular task. An extension to this setting might be useful for certain applications, where exploration strategies that are good for one task (e.g., biking through an environment) are also useful for other tasks (e.g., walking through the same environment).

---

[8]We ran a small number of informal experiments where we allowed BO to select the number of subgoals, but found that `BESD` almost immediately gravitates to the largest number of subgoals (as subgoals come at no cost). Since in the applications that we have in mind, subgoal cost was not a primary concern, we did not pursue this direction as it did not bring any particularly strong insights for the standard case.

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

# A  Proofs

## A.1  Restatement of Theorems

Here we restate the two main theorems from the paper for the reader's convenience. Full proofs are provided in the following sections.

**Theorem 1** *The acquisition function described in ([7](#)) has the property of asymptotic optimality with respect to $\bar{\Theta}$, i.e.,*

$$\lim_{N \to \infty} f(\theta_{rec}^N, \tau_{max}) = \max_{\theta \in \bar{\Theta}} f(\theta, \tau_{max}),$$

*almost surely. That is, the recommended design $\theta_{rec}^N$ becomes optimal as $N \to \infty$.*

**Theorem 2** *The acquisition function of ([7](#)) has bounded asymptotic suboptimality with respect to the original domain $\Theta$ in the sense that with probability at least $1 - \delta$, it holds that*

$$\lim_{N \to \infty} f(\theta_{\mathrm{rec}}^N, \tau_{\max}) \geq \max_{\theta \in \Theta} f(\theta, \tau_{\max}) - d \cdot \|L_\delta\|$$

*where $d = \max_{\theta \in \Theta} \min_{\theta' \in \bar{\Theta}} \mathrm{dist}(\theta, \theta')$ is a measure on the "coarseness" of the discretization and $L_\delta$ is the vector $(L_\delta^1, L_\delta^2, \ldots, L_\delta^m)$, with each $L_\delta^i$ defined as in ([8](#)).*

## A.2  Proof of Theorem 1

The proof is based on theoretical results of Poloczek et al. (2017). Our result, however, includes the ability to select the number of replications $q$. Denote $\lambda(\theta, \tau, q) = \sigma_{\mathrm{env}}^2 + \sigma_{\mathrm{rep}}^2/q$. Also, let $\mathscr{F}^n$ denote the $\sigma$-algebra generated by the history $H^n$. The expectation $\mathbb{E}_n := \mathbb{E}[\,\cdot\,|\mathscr{F}^n]$ is taken with respect to $\mathscr{F}^n$. Recall that $\mu^n$ and $k^n$ are the mean and covariance matrix of the time $n$ belief on $f$. Define the quantities

$$Z^{n+1} = \frac{y^{n+1}(\theta, \tau) - \mu^n(\theta, \tau)}{\sqrt{\mathrm{Var}\big[y^{n+1}(\theta, \tau) - \mu^n(\theta, \tau) \,|\, \mathscr{F}^n\big]}},$$

and

$$\tilde{\sigma}_q^n\big((\theta', \tau'), (\theta, \tau)\big) = \frac{k^n\big((\theta', \tau'), (\theta, \tau)\big)}{\sqrt{\lambda(\theta, \tau, q) + k^n\big((\theta, \tau), (\theta, \tau)\big)}}.$$

Observe that $Z^{n+1}$ is standard normal (conditional on $\mathscr{F}^n$). We have the following recursive updating equation for $\mu^{n+1}$:

$$\mu^{n+1}(\theta, \tau) = \mu^n(\theta, \tau) + \tilde{\sigma}_{q^{n+1}}^n\big((\theta, \tau), (\theta^{n+1}, \tau^{n+1})\big) \, Z^{n+1}, \tag{9}$$

and another recursive formula $k^{n+1}$:

$$\begin{aligned}
k^{n+1}\big((\theta', \tau'), (\theta, \tau)\big) &= k^n\big((\theta', \tau'), (\theta, \tau)\big) \\
&\quad - \tilde{\sigma}_{q^{n+1}}^n\big((\theta', \tau'), (\theta^{n+1}, \tau^{n+1})\big) \big[\tilde{\sigma}_{q^{n+1}}^n\big((\theta, \tau), (\theta^{n+1}, \tau^{n+1})\big)\big]^\top.
\end{aligned} \tag{10}$$

These updating equations are based on the Sherman-Woodbury identity; see Frazier et al. (2009) for a full derivation. The objective of the acquisition function is thus:

$$\begin{aligned}
\frac{\nu^n(\theta, \tau, q)}{q\tau} &= \frac{1}{q\tau} \, \mathbb{E}_n\big[(\mu_*^{n+1} - \mu_*^n) \,|\, (\theta^n, \tau^n, q^n) = (\theta, \tau, q)\big] \\
&= \frac{1}{q\tau} \, \mathbb{E}_n\Big[\max_{\theta'}\big\{\mu^n(\theta', \tau_{\max}) + \tilde{\sigma}_q^n\big((\theta', \tau_{\max}), (\theta, \tau)\big) Z^{n+1}\big\} \\
&\qquad\qquad - \max_{\theta'} \mu^n(\theta', \tau_{\max}) \,\Big|\, (\theta^n, \tau^n, q^n) = (\theta, \tau, q)\Big].
\end{aligned} \tag{11}$$

We also define the quantity

$$V^n(\theta, \tau, \theta', \tau') = \mathbb{E}_n[f(\theta, \tau) \cdot f(\theta', \tau')] = k^n\big((\theta, \tau), (\theta', \tau')\big) + \mu^n(\theta, \tau) \cdot \mu^n(\theta', \tau'). \tag{12}$$

Next, we restate a useful technical lemma from Poloczek et al. (2017).

**Lemma 1 (Restatement of Lemma 1 of (Poloczek et al., 2017))** *Let $\tau, \tau' \in \mathcal{T}$ and $\theta, \theta' \in \Theta$. The limits of the series $\{\mu^n(\theta, \tau)\}_n$ and $\{V^n(\theta, \tau, \theta', \tau')\}_n$ exist. Denote them by $\mu^\infty(\theta, \tau)$ and $V^\infty(\theta, \tau, \theta', \tau')$ respectively. We have*

$$\lim_{n \to \infty} \mu^n(\theta, \tau) = \mu^\infty(\theta, \tau), \tag{13}$$

$$\lim_{n \to \infty} V^n(\theta, \tau, \theta', \tau') = V^\infty(\theta, \tau, \theta', \tau') \tag{14}$$

*almost surely. If $(\theta', \tau')$ is sampled infinitely often, then*

$$\lim_{n \to \infty} V^n(\theta, \tau, \theta', \tau') = \mu^\infty(\theta, \tau) \cdot \mu^\infty(\theta', \tau').$$

Fix a sample path $\omega$, which corresponds to a particular path of measurements and observations

$$\{(\theta^n, \tau^n, q^n, y^{n+1}(\theta^n, \tau^n, q^n))\}_n.$$

By the finiteness of $\bar{\Theta}$, $\mathcal{T}$, and $\mathcal{Q}$, there must exist a configuration $(\theta', \tau', q')$ that is visited infinitely often on sample path $\omega$. The following lemma states the asymptotic behavior of $\nu^n(\theta', \tau', q')/(q'\tau')$ for $n \to \infty$ as a function of $\mu^n(\cdot, \cdot)$ and $\tilde{\sigma}^n_\cdot((\cdot, \cdot), (\cdot, \cdot))$.

**Lemma 2** *Consider the sample path $\omega$ and $(\theta', \tau', q')$ described above. Then, on that sample path $\omega$, it holds that*

$$\lim_{n \to \infty} \tilde{\sigma}^n_{q'}((\theta'', \tau_{max}), (\theta', \tau')) = 0$$

*for every $\theta'' \in \Theta$. Also, the acquisition value tends to zero: $\lim_{n \to \infty} \nu^n(\theta', \tau', q')/(q'\tau') = 0$*

*Proof.* It follows from Lemma 1 that

$$k^n((\theta, \tau), (\theta', \tau')) = \mathbb{E}_n[f(\theta, \tau) \cdot f(\theta', \tau')] - \mu^n(\theta, \tau) \cdot \mu^n(\theta', \tau') \xrightarrow{n \to \infty} 0$$

for any $\theta \in \Theta$, $\tau \in \mathcal{T}$. Then for all $\theta'' \in \bar{\Theta}$, we have

$$\lim_{n \to \infty} \tilde{\sigma}^n_{q'}((\theta'', \tau_{\max}), (\theta', \tau')) = \lim_{n \to \infty} \frac{k^n((\theta'', \tau_{\max}), (\theta', \tau'))}{\sqrt{\lambda(\theta', \tau', q') + k^n((\theta', \tau'), (\theta', \tau'))}} = 0.$$

Note that we made use of the fact that the observation noise $\lambda(\theta', \tau', q') > 0$ for any $q'$. From the proof of Lemma 1 of Poloczek et al. (2017), it is shown that for any $\theta'' \in \bar{\Theta}$,

$$\{\mu^n(\theta'', \tau_{\max})\}_n \quad \text{and} \quad \{\tilde{\sigma}^n_{q'}((\theta'', \tau_{\max}), (\theta', \tau'))\}_n$$

are uniformly integrable (u.i.) families of random variables that converge almost surely to their respective limits $\mu^\infty(\theta'', \tau_{\max})$ and $\tilde{\sigma}^\infty_{q'}((\theta'', \tau_{\max}), (\theta', \tau')) = 0$. Note that the family of random variables $\{\tilde{\sigma}^n_{q'}((\theta'', \tau_{\max}), (\theta', \tau',)) Z^{n+1}\}_n$ is also uniformly integrable since $Z^{n+1}$ is independent of $\tilde{\sigma}^n_{q'}((\theta'', \tau_{\max}), (\theta', \tau'))$. Let $Z$ be a standard normal random variable (independent from all other quantities). It holds that

$$\lim_{n \to \infty} \frac{\nu^n(\theta', \tau', q')}{q'\tau'}$$

$$= \frac{1}{q'\tau'} \Big[ \int_{-\infty}^{+\infty} \phi(Z) \max_{\theta'' \in \bar{\Theta}} \Big\{ \mu^\infty(\theta'', \tau_{\max}) + \tilde{\sigma}^\infty_{q'}((\theta'', \tau_{\max}), (\theta', \tau')) Z \Big\} dZ \tag{15}$$

$$\qquad - \max_{\theta'' \in \bar{\Theta}} \mu^\infty(\theta'', \tau_{\max}) \Big]$$

$$= 0.$$

The first equality is due to (11) and the fact that the operations of summing and taking maximum over a finite set of uniform integrable random variables maintains uniform integrability. ∎

From (7), we know that in each iteration $n$, the configuration $(\theta^n, \tau^n, q^n)$ is selected from according to $\arg\max_{\theta,\tau,q} \nu^n(\theta, \tau, q)/(q\tau)$. Now, for the sake of contradiction, suppose that there exists some configuration $(\breve{\theta}, \breve{\tau}, \breve{q})$ such that $\lim_{n\to\infty} \nu^n(\breve{\theta}, \breve{\tau}, \breve{q})/(\breve{q}\breve{\tau}) > 0$. This immediately leads to a contradiction, since then it cannot be the case that $(\theta', \tau', q')$ is visited infinitely often.

Since the sample path $\omega$ was arbitrary, we conclude that

$$\lim_{n\to\infty} \nu^n(\theta, \tau, q)/(q\tau) = 0 \quad \text{a.s.} \tag{16}$$

for all $\theta \in \bar{\Theta}$, $\tau \in \mathcal{T}$, and $q \in \mathcal{Q}$.

**Lemma 3** *Given that (16) holds, we have that*

$$\arg\max_{\theta\in\bar{\Theta}} \mu^\infty(\theta, \tau_{max}) = \arg\max_{\theta\in\bar{\Theta}} f(\theta, \tau_{max})$$

*almost surely.*

*Proof.* We can conclude from (12) and Lemma 1 that

$$\lim_{n\to\infty} k_n\big((\theta, \tau_{\max}), (\theta, \tau_{\max})\big) = k^\infty\big((\theta, \tau_{\max}), (\theta, \tau_{\max})\big) \quad \text{a.s.}$$

for all $\theta \in \bar{\Theta}$. In the case that the posterior variance $k^\infty((\theta, \tau_{\max}), (\theta, \tau_{\max})) = 0$ for all $\theta \in \bar{\Theta}$, then the maximizer is known perfectly and we are done.

If not, then we define $\hat{\Theta} = \big\{\theta \in \bar{\Theta} \,|\, k^\infty((\theta, \tau_{\max}), (\theta, \tau_{\max})) > 0\big\}$ and consider some $\hat{\theta} \in \hat{\Theta}$ where the posterior variance is positive. Fix any $\hat{q} \in \mathcal{Q}$. We now argue that

$$\tilde{\sigma}_{\hat{q}}^\infty\big((\hat{\theta}, \tau_{\max}), (\hat{\theta}, \tau_{\max})\big) = \tilde{\sigma}_{\hat{q}}^\infty\big((\theta'', \tau_{\max}), (\hat{\theta}, \tau_{\max})\big) \tag{17}$$

for all $\theta'' \in \bar{\Theta}$. Suppose, for the sake of contradiction, that there exist some $\theta_1, \theta_2 \in \bar{\Theta}$ with

$$\tilde{\sigma}_{\hat{q}}^\infty\big((\theta_1, \tau_{\max}), (\hat{\theta}, \tau_{\max})\big) \neq \tilde{\sigma}_{\hat{q}}^\infty\big((\theta_2, \tau_{\max}), (\hat{\theta}, \tau_{\max})\big). \tag{18}$$

Recall (15) and note that it can be rewritten as

$$\lim_{n\to\infty} \frac{\nu^n(\theta', \tau', q')}{q'\tau'} = \frac{1}{q'\tau'}\Big[\mathbb{E}\big[h(Z)\big] - \max_{\theta''\in\bar{\Theta}} \mu^\infty(\theta'', \tau_{\max})\Big], \tag{19}$$

where $h(z) = \max_{\theta''\in\bar{\Theta}}\big\{\mu^\infty(\theta'', \tau_{\max}) + \tilde{\sigma}_{q'}^\infty\big((\theta'', \tau_{\max}), (\theta', \tau')\big) z\big\}$. Since $\bar{\Theta}$ is finite and each function within the maximization in $h$ is affine in $z$, the $h(z)$ is convex[9] and piecewise linear. Since $h$ is convex, there is an affine function $l$ such that

$$l(0) = h(0), \quad l(z) \leq h(z) \text{ for all } z \in \mathbb{R}.$$

The assumption we made in (18), which effectively says that $h$ is created by taking maximum over affine functions of *differing slopes*, implies $h$ cannot itself be affine (and indeed, must consist of various "pieces"). Therefore, there exists an interval $\mathcal{I}$, either of the form $(z_0, \infty)$ or $(-\infty, z_0)$, such that $l(z) < h(z)$ for $z \in \mathcal{I}$. It follows that $\mathbb{E}[l(Z)] < \mathbb{E}[h(Z)]$. By the linearity of $l$, we have

$$\mathbb{E}[l(Z)] = l(\mathbb{E}[Z]) = l(0) = h(0) = \max_{\theta''\in\bar{\Theta}} \mu^\infty(\theta'', \tau_{\max}) < \mathbb{E}\big[h(Z)\big].$$

This implies that (19) is strictly positive, contradicting (16). We thus conclude that (17) holds, which is equivalent to

$$\frac{k^\infty\big((\theta'', \tau_{\max}), (\hat{\theta}, \tau_{\max})\big)}{\sqrt{\lambda(\hat{\theta}, \tau_{\max}, \hat{q}) + k^\infty\big((\hat{\theta}, \tau_{\max}), (\hat{\theta}, \tau_{\max})\big)}} = \frac{k^\infty\big((\theta''', \tau_{\max}), (\hat{\theta}, \tau_{\max})\big)}{\sqrt{\lambda(\hat{\theta}, \tau_{\max}, \hat{q}) + k^\infty\big((\hat{\theta}, \tau_{\max}), (\hat{\theta}, \tau_{\max})\big)}},$$

---

[9]Pointwise maximum of convex functions is convex.

for all $\theta'', \theta''' \in \bar{\Theta}$. Moreover, since $\hat{\theta}$ was chosen from $\hat{\Theta}$, we know that

$$\lambda(\hat{\theta}, \tau_{\max}, \hat{q}) + k^{\infty}\big((\hat{\theta}, \tau_{\max}), (\hat{\theta}, \tau_{\max})\big) > 0,$$

and hence $k^{\infty}\big((\theta''', \tau_{\max}), (\hat{\theta}, \tau_{\max})\big) = k^{\infty}\big((\theta'', \tau_{\max}), (\hat{\theta}, \tau_{\max})\big)$ for all $\theta'', \theta''' \in \bar{\Theta}$.

This means the covariance matrix of $\{f(\theta, \tau_{\max}) \,|\, \theta \in \bar{\Theta}\}$ is proportional to the all-ones matrix, and that draws from $f(\theta, \tau_{\max}) - \mu^{(\infty)}(\theta, \tau_{\max})$ are *constant* across $\theta \in \bar{\Theta}$. Therefore, $\arg\max_{\theta \in \bar{\Theta}} \mu^{(\infty)}(\theta, \tau_{\max}) = \arg\max_{\theta \in \bar{\Theta}} f(\theta, \tau_{\max})$ and the statement of the theorem holds. ∎

### A.3 Proof of Theorem 2

In Theorem 2, we establish an additive bound on the loss of the solution obtained by BESD, $f(\bar{\theta}, \tau_{\max})$, with respect to the unknown optimum $f(\theta^{\mathrm{OPT}}, \tau_{\max})$, as the number of iterations $N \to \infty$. Recall that we suppose $\mu(\theta, \tau) = 0$ for all $\theta, \tau$, and that the kernel $k(\cdot, \cdot)$ has continuous partial derivatives up to the fourth order. According to Theorem 3.2 of Lederer et al. (2019), for any $\delta \in (0, 1]$, with probability at least $1 - \delta$, the quantity

$$\|L_\delta\| = \big\|(L_\delta^1, L_\delta^2, \cdots, L_\delta^m)\big\|$$

is a Lipschitz constant of $f$ on $\Theta$, i.e., it holds that

$$|f(\theta, \tau_{\max}) - f(\theta', \tau_{\max})| \le \|L_\delta\| \cdot \mathrm{dist}(\theta, \theta'),$$

where $\theta, \theta' \in \Theta$. By the definition of $d$, there exists a $\bar{\theta} \in \bar{\Theta}$ such that $\mathrm{dist}(\bar{\theta}, \theta^{\mathrm{OPT}}) \le d$. Therefore, it follows that the suboptimality due to optimizing in $\bar{\Theta}$ is bounded by

$$f(\theta^{\mathrm{OPT}}, \tau_{\max}) - f(\bar{\theta}, \tau_{\max}) \le \|L_\delta\| \cdot d. \tag{20}$$

Theorem 1 completes the proof of Theorem 2 since (20) holds with probability $1 - \delta$.

## B  GP Hyperparameter Estimation

The hyperparameters of the covariance function $k$ are set via *maximum a posteriori* (MAP) estimation. Recall that a MAP estimate is the mode under the log-posterior obtained as the sum of the log-marginal likelihood of the observations and the logarithm of the probability under a hyper-prior. We focus on describing the hyper-prior, since the log-marginal likelihood follows canonically; see (Rasmussen & Williams, 2006, Ch. 5) for details. The proposed prior extends the hyper-prior for the multi-task GP model used in Poloczek et al. (2017). We set the mean function $\mu$ and the noise function $\lambda$ to constants that we estimate. For the covariance function we need to estimate $d + 5$ hyperparameters: the signal variance, one length scale for every subgoal parameter in $\theta$ and the four parameters associated with $k_\tau$. We suppose a normal prior for these parameters. For the signal variance, the prior mean is given by the variance of the observations, after subtracting the above estimate for the observational noise. Here we use the independence of observational noise that we argued in Section 3.5. For any length scale, we set the prior mean to the size of the interval that the associated parameter is chosen in. Having determined a prior mean $\mu_\psi$ for each hyperparameter $\psi$, we may then set the variance of the normal prior to $\sigma_\psi^2 = (\mu_\psi/2)^2$.

