# OpenReview forum: "Dynamic Subgoal-based Exploration via Bayesian Optimization"
_TMLR — Accepted by TMLR_

### Review · Reviewer_1jSM · 2023-04-14

**Summary Of Contributions:**

This paper aims to improve the sample efficiency of reinforcement learning (RL) algorithms in settings with sparse rewards and costly environment interactions. Specifically, the authors introduce a framework for learning exploration policies that employ a sequence of subgoals, which are trained in a few-shot Bayesian meta optimization setting. The training tasks are drawn from a distribution over parameterized MDPs, and the evaluation is performed on separate tasks sampled from the same distribution. The subgoals are structured by the duration and number of episodes, with the ultimate goal of improving average performance across the task distribution, particularly in spatial navigation tasks. The authors suggest that their algorithm provides useful subgoal structures to exploration policies.

Experiments show a mix of qualitative and quantitative support. The authors evaluate the proposed algorithm on a number of grid world problems and the Mountain Car domain. They provide visualizations of several samples of subgoal sequences and argue that their location and ordering make intuitive sense for the domain at hand. While some examples demonstrate reasonable subgoal sequences, others appear less helpful. The quantitative results are presented in the form of plots showing that the quality of the solution improves as the algorithm is provided with more data. The plots also demonstrate comparatively better trends than the chosen baselines. Although the presentation of the data could be clearer, especially with a different choice of baselines, the results overall show generally positive trends.



**Audience:**

Yes

**Broader Impact Concerns:**

This paper does not present any ethical concerns that would require a broader impact statement.

**Claims And Evidence:**

Yes

**Requested Changes:**

- Comment on how this work brings RL closer to real-world settings, as your motivation suggests it can.
- Provide a more candid discussion of related work.
- Section 3.1: Change $\gamma \in [0,1]$ to $\gamma \in [0,1)$ .
- Section 3.1: Change "arbitrary distribution" to be something more specific.
- Section 3.1: Define what a sparse reward is precisely.
- Section 3.1: Define the $\epsilon$ -greedy policy.
- Equation 1: Make it clear that $a_t\sim \pi(\cdot|s_t)$ and $s_1=s$ .
- Section 3.2: Define a potential function in this context.
- Equation 2: Make it clear that this holds for all $s,s'\in\mathcal{S}$
- Example 1: Define $n$ .
- Describe the exploration policy you consider.
- Describe how the subgoals inform the exploration policy. the process how the subgoals.
- Section 4.1: Justify the choice of kernels and their hyperparameters.
- Algorithm 1: Describe how you estimate hyperparameters of the GP prior.
- Include a section that introduces Bayesian Optimization.
- Section 4.2: Introduce the notion of Bayes optimality.
- Include a proof of Proposition 1.
- Include a summary of the methodology followed in each experiment.
- Include results from another random baseline that employs experience from multiple randomly selected environments.
- Remove comparison to MAML.
- Report single standard error bars (not double standard error).

**Strengths And Weaknesses:**

### Strengths
- The question of how subgoals should and can shape the structure of exploratory behavior is interesting.
- Findings from this research could be potentially impactful---by exemplifying how to improve sample efficiency in large domains with non-Markovian dynamics.
- The study includes reasonable choices for evaluation domains.
- The study uses a reasonable number of domains.
### Weaknesses
- Presentation of technical concepts makes this paper difficult to follow.
- Originality of some theoretical results is unclear.
- The theoretical results have limited significance in the context of this study.
- Empirical methodology is unclear.
- Empirical support could be stronger.

Further comments are provided below.

### General Comments

- Many technical concepts are used before they are defined. I provide a few examples, but there are more in the text.
  - Introduction: The notion of subgoals should be described before the final paragraph of the introduction.
  - Section 3.2: artificial reward
  - Section 3.2: intrinsic reward
  - Section 4: acquisition function
  - "A dynamic subgoal strategy is an ordered set of subgoals (along with associated rewards leading to each subgoal, omitted here for illustrative clarity) that leads the agent on a trajectory where the underlying RL algorithm is more likely to discover the optimal behavior."
    - There is a lot of unpack here, especially early without any additional context in the introduction.
    - Define a subgoal.
    - Define a subgoal strategy.
    - Describe what it means for a subgoal strategy to be "dynamic".
- The related work seems somewhat superficial, and it appears to have been written with the intention of avoiding any potential challenges to the contributions of this paper. I would suggest that the paper could benefit from a more thorough and candid discussion of related work. For instance, the paper claims the work of Gupta et al. 2018 is "impractical" without supporting evidence of why.
- Be careful with overstating your claims. For instance, "Theta is an arbitrary distribution." Certainly this to respect some constraints to remain relevant to your problem setting.
- Mention where the proofs can be found before stating the formal claims.
- Several references should be enclosed in parentheses. For example, Page 12: "Clark et al. (2014)" should be in parentheses.
- Colors in your figures need to be bolder.
- This statement is not true in general: "For example, in a 20 × 20 gridworld with a sparse reward, the goal is not even reached for the first time by a standard Q-learning agent (let alone find an optimal policy) after 10 million interactions."
- This is vague: "if properly designed, can direct the agent to explore useful parts of the state space." Be specific about what is meant by "useful" and "properly designed."
- Section 3.2: What is a subgoal parameterization?
- Example 1: I took a while to understand this example, because it references a figure on page 1, when the text is on page 6.
- It appears that, in the examples you provided, the sequence of subgoals is just as critical, if not more so, than the actual location of the goals. Could you please elaborate on how your method ensures the correct sequencing of these subgoals?
- "the dimension of $\theta$ only needs to scale with the number components of s that pertain to the spatial positioning of the agent."
  - This suggests that the applicability of your contributions apply to only spatial settings. Can you expand on this potential limitation?
- Section 3.3: It is worth noting how your "subgoal-augmented MDP" is a factored MDP.
- Figure 3a: The colors are very faint.
- Section 3.4: Does the training budget $\tau$ include steps over multiple episodes, or is this the maximum length of a single episode?
- Section 3.4: Referring to a generic RL algorithm is too vague. This needs to be restricted to one that explores in a certain way, no?
- Equation 4: It is not clear where $i_0$ is used.
- Section 3.5: The total training cost is not necessarily proportional to $\tau^n$, since any given episode could take fewer steps.
- Section 4.1: The performance of the algorithm is assumed to be i.i.d according to the GP prior. However, the performance of subsequent policies benefit from prior experiences. Comment on how the significance of this assumption violation.
- Equation 7: It is a bit confusing to call this the output of a policy and an action the output of a policy.
- The theoretical arguments seem to contain missing assumptions, and its proofs are also missing steps that could be helpful to the reader. For example, assumptions made about convexity (Appendix A).


### Experiment Comments
- Differences from the Q-learning baseline quantify the degree to which subgoals are useful. It may help to call this this baseline "No Subgoals".
- The Transfer Q-learning is initialized with experience from one randomly selected training environment.
  - I understand that this was meant to be a random baseline. However, unlike the other baselines, it does not benefit from experience gained in multiple environment instances. I believe that it would be beneficial to include another baseline that employs experience from multiple randomly selected environments, essentially training sequentially on a random sequence of environments. This baseline is expected to perform worse than your proposed method, assuming that the learned subgoal sequences are useful.- Performance differences from the adaptive configuration baseline are unclear.
- I have the impression that the MAML baseline is somewhat of a straw man comparison, given that Bayesian Optimization is typically utilized in scenarios where gradients are unavailable. Indeed, the outcomes indicate that MAML underperforms the proposed method, though its performance has not always flatlined. Nevertheless, I do not believe you need to compare with MAML to prove your point. In fact, applying MAML to the subgoal learning problem seems to be a topic that warrants a distinct line of inquiry.
- " in order to bypass the wall,"
  - I believe that it may be inaccurate to assert that the algorithm produced this output for a specific reason. It is more appropriate to say that this was the best possible output under the given constraints and information.
- Section 5.2.1: I think the take away here is generally correct: When structuring the exploration strategy with a sequence of two subgoals, the subgoals appear to be located in places one would expect. However, I think this point could be made more clearly with cleaner figures and some supporting text.
- Section 5.4.1: It is unclear what Behavior B is intended to illustrate. This seems to be a negative result.
- Section 5.5: This feels like an environment that could have benefited from more than two subgoals. It would be compelling if you were able to show an oscillatory pattern in the direction the subgoals take the agent.
- In this section, the paper first presents several baselines before evaluating the qualitative aspect of a single method (BESD). To enhance readability, it may be helpful to postpone the introduction of these baselines until Section 5.7. Another option could be to split this section up by each experiment rather than by the evaluation domain.
- Figure 7: The plots presented do not effectively convey the point that the proposed subgoal structure leads to an enhancement in sample efficiency when averaged over the test environments.
- Instead of total cost, it would be more clear report the number of environment interactions as the performance measure.
- Table 1: It is not clear what $\tau$ means here. Is it for a single value of $m$ ?
- Table 1: I'm confused by what to take away from this.
- The domains used here are reasonable. However, they are still solvable with standard approaches.  Other domains, where interaction costs are prohibitively high for standard exploration approaches, would provide stronger support for your main claim.

### Other Questions
- What is a "Latin Hypercube"?
- What does it mean for a subgoal to be complete?
- Figure 1: Why are the subgoal locations not centered on the cells? Is this significant?
- Section 3.5: Is $\mathcal{T}$ just the set of natural numbers?
- Does $q^n$ control the number of episodes?
- Section 5.2.1: What were the final values selected for the number of episodes and the steps per episode?
- Figure 5b: What does the vertical axis represent?
- Shouldn't the last subgoal always be the original goal?

---

### Review · Reviewer_LUqN · 2023-04-16

**Summary Of Contributions:**

The manuscript presents a "cost-efficient" framework for learning dynamic subgoal exploration strategies in reinforcement learning environments characterized by sparse rewards and expensive interactions. The authors leverage Bayesian optimization to develop the Bayesian Exploratory Subgoal Design (BESD) algorithm, which learns a set of useful subgoals for a distribution of environments under a couple of constraints. The proposed approach is thoroughly evaluated on various grid worlds against a meta-learning baseline and a few BO algorithms.


**Audience:**

Yes

**Claims And Evidence:**

Yes

**Requested Changes:**

1. I would appreciate a tightening of scope of the manuscript to either (1) be extremely clear that this method is currently relevant primarily to a particular kind navigation domain with very well defined constraints, and/or (2) discuss the limitations of the method wrt. these domain constraints.
2. It would be beneficial to discuss the differences and connections between the proposed method and options learning in HRL. Are there any unique aspects to the presented method that distinguish it from existing HRL approaches? Addressing this question will help to further clarify the paper's novelty and contributions.
3. It would be useful to provide a more detailed comparison between BESD and such methods, addressing this point. How does the proposed method handle the cost of exploration differently, and what advantages does this bring? By clarifying these aspects, the authors can better position their work within the context of existing approaches.

By addressing these additional points, the authors can further strengthen their paper and provide a more comprehensive understanding of the proposed method's advantages and limitations compared to existing approaches in the literature.

**Strengths And Weaknesses:**

### Strengths

1. The paper addresses a relevant and important problem, which has been underexplored in the literature: cost-efficient exploration strategies for RL environments with expensive interactions and sparse rewards.
2. The proposed BESD algorithm is well-motivated and grounded. The manuscript does an excellent job at describing it, using the right level of formalism as well as being clear in design choices for all of its components. Well done!
3. The experimental evaluation is comprehensive wrt. the chosen testbeds, providing a clear comparison between the proposed method and existing baseline approaches.
4. Overall, the paper is well-written and structured, making it easy to understand the proposed method and its contributions.

### Weaknesses

1. The experiments are mainly limited to grid world environments, raising questions about the method's performance in more complex or real-world settings.
2. It feels like the manuscript is trying to argue that BESD is a generic method for discovering efficient exploration "policies", however to me it seems fairly clear that it might be limited to environments where (a) it is possible to define state based goals, (b) it is possible to define a potential function for each subgoal. The manuscript sometimes tries to clarify that this method is really targeted at navigation tasks (which makes a lot of sense -- it is well suited to them!), but overall it feels like it's a bit of a misdirection that is not very warranted.
3. The problem formulation presented in this paper bears some resemblance to options learning. One might easily summarize the method as effectively learning a set of environment-distribution specific options, using BO.
4. The authors argue that BESD is superior to "standard" intrinsic exploration RL methods because it considers the cost of exploration. However, it is important to note that the cost of exploration is often marginalized away when the exploration strategy is trained alongside the learned policy in intrinsic exploration methods.

---

### Review · Reviewer_k6VU · 2023-04-22

**Summary Of Contributions:**

The paper considers the problem of policy optimization in environments where interacting with the environment is expensive, thus only allowing for a limited number of interactions. The experiments in the paper considers an agent that faces a distribution of tasks, experienced through a number of training environments. The agent from its training environments must learn an exploration strategy so that when it encounters a new environment, it can use this strategy to improve its performance and learning. The paper presents a cost-aware Bayesian optimization approach to search over  a class of subgoal-based exploration strategies. The experiments show that the proposed algorithm relatively improves existing meta-learning algorithms like MAML and Hyperband.

**Audience:**

Yes

**Broader Impact Concerns:**

A broader impact statement was not available.

**Claims And Evidence:**

Yes

**Requested Changes:**

* To strengthen the contribution, it would be useful to see how the approach scales when there are more than 2-3 goals in the domain, along with comparisons with relevant baselines.
* Also adding a discussion on how the approach can scale to much more complex domains will be particularly useful to the community.
* Another direction could be to improve the baselines: compare against MAML with its optimal hyperparameters and also include simpler non-Bayesian hierarchical RL baselines that seem natural for the tasks considered.
* Finally, it would also be interesting to compare against the different trajectories generated by the baselines in one of the tasks.


**Strengths And Weaknesses:**

Strengths:
* Paper is well-written, the algorithm is simple to understand.
* Related work section is exhaustive and clearly describes relevant prior work.
* Section 3 (Problem formulation) is particularly useful and sets up the task setup that is considered in their method and experiments.
* The different subgoal trajectories selected by their approach are presented in Figure 4, adding more clarity in how the agent operates.
* Empirical section compares the proposed method against a number of baselines

Weaknesses:
* The domains in the experiments seem to be limited; only 2 or 3 goal selections are necessary in these domains.
* The experiments do not seem to make a fair comparison with MAML. In page 13, it is mentioned that the batch size in outer-loop is set to 1. This seems like a severe limitation. What happens to the experiments if MAML was used with a reasonable choice of hyperparameters (perhaps as defined in its paper)?
* Another simple approach could be a hierarchical RL approach where the higher level selects goals and the lower level selects actions conditioned on the goal and current state. The experiments could be more insightful if it included such a simple baseline.
* A discussion on how the presented approach can be scaled up to much more complex environments (with potentially 1000s of goals) is missing.

---

### Decision · Action_Editors · 2023-05-29

**Recommendation:** Accept with minor revision

**Comment:**

All 3 reviewers agreed on acceptance. There have been major improvements to the paper due to the authors responding well to the reviews and the extensive interactions with the reviewers.

I understand the authors are still waiting on new results. Once those are ready I will make a final check of the paper.

**Audience:**

The papers focus on domains where interaction is limited due to expensive interactions with the world is very relevant for both applications and algorithm researchers alike. The paper is relevant to those working in various forms of meta-RL and never-ending RL.

**Claims And Evidence:**

This paper nicely achieves the stated goals: introducing a new meta-algorithm that produces a sequence of subgoals in the limited environment-interaction setting with the goal of solving a tasks from a distribution. The paper provides reasonable theory and a nice set of experiments in small domains with both qualitative and quantitive analysis, many baselines and good statistical practices (50 independent runs!).